# Exploring the effect of training set size and number of categories on ice crystal classification through a contrastive semi-supervised learning algorithm

Yunpei Chu[1,2], Huiying Zhang[1], Xia Li[3], and Jan Henneberger[1]

[1]Institute for Atmospheric and Climate Science, ETH Zurich, Zurich, Switzerland
[2]Department of Geosciences and Remote Sensing, Faculty of Civil Engineering and Geosciences, Delft University of Technology, Delft, Netherlands
[3]Institute for Machine Learning, ETH Zurich, Zurich, Switzerland

**Abstract.** The shapes of ice crystals play an important role in global precipitation formation and radiation budget. Classifying ice crystal shapes can improve our understanding of in-cloud conditions and these processes. Existing classification methods rely on features such as the aspect ratio of ice crystals, environmental temperature, and so on, which bring high instability to the classification performance, or employ supervised machine learning algorithms that heavily rely on human labeling. This poses significant challenges, including human subjectivity in classification and a substantial labor cost in manual labeling. In addition, previous deep learning algorithms for ice crystal classification are often trained and evaluated on datasets with varying sizes and classification schemes, each with distinct criteria and a different number of categories, making it difficult to make a fair comparison of algorithm performance. To overcome these limitations, a contrastive semi-supervised learning (CSSL) algorithm for the classification of ice crystals is proposed. The algorithm consists of an upstream unsupervised learning network tasked with extracting meaningful representations from a large amount of unlabeled ice crystal images, and a downstream supervised network is fine-tuned with a small subset labeled images of the entire dataset to perform the classification task. To determine the minimal number of ice crystal images that require human labeling while maintaining the algorithm performance, the algorithm is trained and evaluated on datasets with varying sizes and numbers of categories. The ice crystal data used in this study was collected during the NASCENT campaign at Ny-Ålesund and CLOUDLAB project on the Swiss plateau, using a holographic imager mounted on a tethered balloon system. In general, the CSSL algorithm outperforms a purely supervised algorithm in classifying 19 categories. Approximately 154 hours of manual labeling can be avoided using just 11 % (2048 images) of the training set for fine-tuning, sacrificing only 3.8 % in overall precision compared to a fully supervised model trained on the entire dataset. In the 4-category classification task, the CSSL algorithm also outperforms the purely supervised algorithm. When fine-tuned on just 2048 images (25 % of the dataset), it achieves an overall accuracy of 89.6 % accuracy-nearly matching the 91.0 % accuracy of the supervised algorithm trained on 8192 images. When tested on the unseen CLOUDLAB dataset, CSSL shows superior generalization, improving accuracy by an average of 2.19 %. Our analysis also reveals that both CSSL and purely supervised algorithms exhibit inherent instability when trained on small dataset sizes, as well as the performance difference between them converges as the training set size exceeds 2048 samples.These results highlight the strength and

practical effectiveness of CSSL in comparison to purely supervised methods and the potential of the CSSL algorithm to perform
well on datasets that would be collected under different conditions.

## 1  Introduction

The shape of the ice crystals has a strong influence on the radiative properties of ice and mixed-phase clouds. It mainly affects
the scattering and absorption properties of clouds (Wendling et al., 1979; Yang et al., 2000). Considering different shapes
in the climate model, both long-wave terrestrial radiative forcing and short-wave radiative forcing of the Earth can change
significantly (Wendisch et al., 2007; Liou et al., 2008; Järvinen et al., 2018). In addition, global precipitation is influenced by the
shape of the ice crystals. The intensity, duration, and type of precipitation are governed by the shapes of the ice crystals within
the cloud and the associated microphysical processes and environmental conditions (Thériault et al., 2012; van Diedenhoven
et al., 2014; Santachiara et al., 2016; Schlenczek et al., 2017; Gupta et al., 2023).
Different environmental conditions and complex microphysical processes in clouds result in a wide variety of ice crystal
shapes found in nature. Ice crystals grow into basic habits such as columns, plates, and dendrites under different temperatures
and supersaturation (Lohmann et al., 2016). Through depositional growth, basic habits accumulate mass and eventually fall
under gravity. They collide with supercooled droplets and other ice crystals as a result of the difference in the fall speed. This
leads to two microphysical processes: riming and aggregation. In aggregation, two ice crystals stick together, and in riming,
supercooled droplets freeze onto the ice crystals. Both processes change the shape of the ice crystals (Lohmann et al., 2016).
The shapes of ice crystals can reveal the ambient conditions and microphysical processes in which they grew (Pasquier et al.,
2023). Therefore, it is essential to have a classification algorithm for ice crystals.

With the development of airborne single particle imaging probes, much research focused on classifying ice crystals based
on the geometric and environmental features of the ice crystal such as the maximum diameter, aspect ratio, and environmental
temperature, using machine learning methods such as principal component analysis (Lindqvist et al., 2012) and multivariate
logistic regression (Praz et al., 2018). Deep learning-based methods for the classification of ice crystals began to emerge when
Xiao et al. (2019) proposed the use of convolutional neural networks (CNNs) to classify ice crystal images taken by the Cloud
Particle Imager (CPI). CNN is a popular network structure used in various fields of atmospheric science for different visual
learning tasks such as cloud image classification (Ye et al., 2017; Lv et al., 2022), and semantic segmentation for ground-
based cloud images (Song et al., 2020). Recently, Schmitt et al. (2024) applied Visual Geometry Group (VGG) algorithm
which is a kind of CNN to classify scattering pattern images of ice crystals. However, Xiao et al. (2019) and Schmitt et al.
(2024)'s algorithm was limited to single-label classification and therefore was not capable of classifying complex ice crystals
that undergo microphysical processes. Zhang et al. (2022) proposed a multi-label scheme that considers both basic habits and

microphysical processes, and Zhang et al. (2024) developed a rotated object detection algorithm for ice crystals that can classify the ice crystal down to the aggregated ice monomer scale.

Although deep learning-based methods can automatically extract more detailed features (Li et al., 2021), current methods were all based on supervised learning that is highly dependent on extensive manual labeling for training. Experts have a time-consuming and strenuous job labeling ice crystal images (Xiao et al., 2019; Zhang, 2021; Jaffeux et al., 2022; Zhang et al., 2024). Furthermore, we cannot make a fair comparison of the models proposed (Lindqvist et al., 2012; Praz et al., 2018; Xiao et al., 2019; Schmitt et al., 2024; Zhang et al., 2024) so far because they were trained and tested with datasets of different numbers of categories and sizes, but the size and number of categories of training sets are important factors affecting the classification accuracy of the models.

To overcome the limitations, we propose a contrastive semi-supervised learning algorithm (CSSL). The algorithm consists of two main stages: unsupervised pre-training in the upstream network and supervised fine-tuning in the downstream network. During the unsupervised pre-training phase, the algorithm extracts features from a large collection of unlabeled images of ice crystals and learns the specific features of different shapes of ice crystals. These learned features are then transferred to the downstream network by inheriting the weights of the upstream network during the supervised fine-tuning stage. By using the pre-learned features, the need for labeled images in the fine-tuning process is significantly reduced, leading to improved efficiency and performance. The algorithm aims to reduce the workload of manual labeling ice crystal images when training a deep-learning based model while achieving a higher classification accuracy compared to traditional fully supervised learning algorithms. In addition, we explore the influence of two factors: 1. the size of training set and 2. the number of categories in the classification performance. Understanding how classification accuracy is affected by the size of the training set will be valuable for researchers working with limited labeled datasets. The data used in this study are described in Sect. 2. The detailed introduction to CSSL is given in Sect. 3. The implementation is presented in Sect. 4. Section 5 presents the results and discussions of our study. Section 6 presents the conclusions and relevant outlooks.

## 2 Data

This study used three ice crystal image datasets to train and test. These images were collected during the NASCENT campaign, which was conducted in Ny-Ålesund, Svalbard, Norway (Pasquier et al., 2022a) and during CLOUDLAB project which is conducted on Swiss Plateau in center Switzerland (Henneberger et al., 2023). The ice images were collected by HOLographic Imager for Microscopic Objects 3B (HOLIMO3B), which is a holographic imager carried by a tethered balloon system (Ramelli et al., 2020). A summary of datasets used in this study is shown in Table 1.

The original dataset includes both ice crystals, cloud droplets and artifacts. Ice crystals were distinguished from the rest particles in the dataset using CNN, and details can be found in Touloupas et al. (2020). The images of ice crystals were then manually labeled according to the multi-label classification scheme described in (Zhang et al., 2024). The dataset includes 19 categories in NASCENT19, 14 categories in NASCENT20 and 4 categories in CLOUDLAB, with the categories in NASCENT20 and CLOUDLAB being the subset of those in NASCENT19 (Table 1).

| Dataset | Instrument | Location | Time | Number | Categories |
|---------|-----------|----------|------|--------|-----------|
| NASCENT19 | HOLIMO3B | Ny-Ålesund, Norway | November 2019 | 18,864 | 19 |
| NASCENT20 | HOLIMO3B | Ny-Ålesund, Norway | April, 2020 | 14,490 | 14 |
| CLOUDLAB | HOLIMO3B | Eriswil, Switzerland | January 2023 | 2,143 | 4 |

**Table 1.** All datasets were collected by the same instrument. The number in the fifth column indicates the number of labeled images. NASCENT19/20 were labeled by (Zhang et al., 2024). The CLOUDLAB was labeled by the authors (see Author Contributions). The difference in collection date and location leads to different number of categories showing in the last column.

The classification scheme in this study follows the one proposed in (Zhang et al., 2024), taking into account both basic habits and microphysical processes. There are 19 categories in the scheme (Table 2). The scheme contains 7 basic habits identified by Pasquier et al. (2022b) from the ice crystal images collected and identified in mixed-phase clouds of Ny-Ålesund during the NASCENT campaign (Pasquier et al., 2022a). When combined with two microphysical processes: aging and aggregation, these basic habits develop into 12 complex shape categories, after excluding combinations that were not feasible. Among the 7 basic habits, the "Plate" and "Column" formed due to deposition growth under different temperature and supersaturation conditions. "Lollipop" (Keppas et al., 2017) forms by a droplet freezing on a columnar ice crystal, or the columnar part is the result of depositional growth on a frozen droplet. "CPC (columns on capped-columns)" originated from cycling through the columnar and plate temperature growth regimes, during their vertical transport by in-cloud circulation (Pasquier et al., 2023). Ice crystals that are too small for shape determination are categorized as "Small", while large crystals with indistinguishable shapes are categorized as "Irregular". As for the two microphysical processes, "aggregate" is the ice crystals with different basic habits sticking together. If there were more than two individual basic habits identified in one image, it will be tagged as "aggregate". In images we can observe sharp edges and clear individual components of basic habits that undergo depositional growth. "Aged" indicates that the ice crystals undergo processes such as riming, melting or sublimation. For example, in the case of riming, the images of aged crystals usually show softly textured edges, which represent the supercooled droplets freezing on them.

The three datasets were collected under different environmental conditions in the cloud, which are caused by different weather conditions (Pasquier et al., 2022b; Henneberger et al., 2023), and therefore the distributions of ice crystal categories in these datasets are very different. Figure 1 shows the distribution of categories in the three datasets and the example images for each category. The ice crystals in NASCENT19 were dominated by "column", "CPC" and "column_aged", which accounted for 66.5 % in total, while the dominated categories in NASCENT20 were mainly "irregular", "small" and "irregular_aggregate", accounting for 54.4 %. In addition, "lollipop", "CPC", "lollipop_aggregate", "droplet_aggregate" and "CPC_aggregate" were not detected in NASCENT20. The CLOUDLAB dataset has 2143 labeled images used in this study. These images have 4 categories and are all column-related shapes including "column", "column_aged", "column_aggregate" and "column_aged_aggregate". The categories present in NASCENT20 and CLOUDLAB are indicated in Table 2. The large discrepancy between the distribution of categories in the three datasets demonstrates the natural variability of ice crystal shapes in different observations

| | Class | Description | NASCENT20 | CLOUDLAB |
|---|---|---|---|---|
| Basic habits | Column | Columnar ice crystals | Yes | Yes |
| | Plate | Plate-like ice crystals | Yes | No |
| | Lollipop | A lollipop forms when a single drizzle-sized water droplet collides with a single columnar ice crystal and freeze on it. | No | No |
| | Frozen droplet | A supercooled water droplet freezes due to the presence of Ice nuclei particles (INP) in it or contacting with an INP. It usually has a non-spherical shape. | Yes | No |
| | Irregular | Irregular-shaped ice crystals that cannot be clearly defined as any ice basic habit. | Yes | No |
| | Small | Ice crystals that are too limited in pixel number to accurately determine their basic habit (usually smaller than 75 $\mu m$) | Yes | No |
| | CPC | Ice crystals with columns on capped-columns (CPC) formed when growing under both column and plate temperature conditions (Pasquier et al., 2023). | No | No |
| Complex shapes | Column_aged | Columnar ice crystals that have supercooled cloud droplets frozen on different faces of them. | Yes | Yes |
| | Column_aggregate | Columnar ice crystals that aggregate with basic habits (including Columns). | Yes | Yes |
| | Column_aged_aggregate | Columnar ice crystals that undergo both aggregation and aging. | Yes | Yes |
| | Plate_aged | Plate-like ice crystals that have supercooled cloud droplets frozen on different faces of them. | Yes | No |
| | Plate_aggregate | Plate-like ice crystals that aggregate with basic habits (including Plates). | Yes | No |
| | Plate_aged_aggregate | Plate-like ice crystals that undergo both aggregation and aging. | Yes | No |
| | Lollipop_aggregate | Lollipop ice crystals aggregate with basic habits (including Lollipops). | No | No |
| | Droplet_aged | Frozen droplets that undergo aging and have supercooled cloud droplets riming around it. | Yes | No |
| | Droplet_aggregate | Frozen droplets that aggregate with basic habits (including Frozen droplets). | No | No |
| | Droplet_aged_aggregate | Frozen droplets that undergo both aggregation and aging. | Yes | No |
| | Irregular_aggregate | Irregular-shaped ice crystals aggregate with basic habits (including Irregulars). | Yes | No |
| | CPC_aggregate | CPC aggregate with other habits (including CPCs). | No | No |

**Table 2.** Description of 19 categories of ice crystals according to the classification scheme proposed in (Zhang et al., 2022). It includes 7 basic habit categories and 12 more complex categories. The last two columns of the table record whether the category is included by NASCENT20 and CLOUDLAB. Modified from (Zhang et al., 2024)

(Zhang et al., 2024). According to Pasquier et al. (2022a, b), NASCENT19 was collected when the temperature in the cloud ranged from $-8°C$ to $-1°C$ while NASCENT20 was collected when the temperature in the cloud was between $-22°C$ and $-15°C$. These temperatures indicate that the ice crystals in NASCENT19 formed in the column temperature regime while the ice crystals in NASCENT20 formed in the plate temperature regime Lamb and Verlinde (2011). Those large particles such as the "irregular" and "irregular_aggregate" likely collided with each other during the sediment from the cloud aloft (Pasquier et al., 2022b). The images in CLOUDLAB were collected during the seeding experiments inside the supercooled stratus clouds. Experiments and approaches were described in Henneberger et al. (2023). The in-cloud temperature ranges from $-8°C$ to $0°C$, which is temperature regime for column growth. Therefore, the data used from CLOUDLAB are dominated by column related shapes. A comprehensive collection of ice crystal examples can be found in the appendix of Zhang et al. (2024), where images of each distinct category are presented with scale bars indicating their actual dimensions.

## 3 Method

This study applied a contrastive semi-supervised learning (CSSL) algorithm to classify ice crystal images. It consists of two parts of neural networks (Fig. 2). The upstream network is an unsupervised contrastive pre-training that aims to learn useful features of images without human supervision (i.e. image labels) using a pretext task (Zhang et al., 2017). Such useful features can be easily adapted to the downstream network for specific tasks such as image classification. A common pretext task in contrastive learning is instance discrimination (Wu et al., 2018; He et al., 2020; Chen et al., 2020). Instance discrimination is based on the concept that the features of images in a latent space, or known as embeddings, should reflect their visual similarity. Specifically, embeddings of visually similar images should be positioned closer together in the latent space, while those of visually dissimilar images should be further apart. This principle ensures that the model can effectively differentiate between instances by clustering similar images and separating dissimilar ones based on their visual characteristics. In the context of ice crystal classification, the general aim of unsupervised contrastive learning is to move ice crystal images of different categories away from each other in latent space while bringing images of the same categories as close together as possible, without knowing the labels of images. The upstream network has three stages: Stage 1 is data augmentation, Stage 2 is the encoder and projector, and Stage 3 is the calculation of the similarity (Fig. 2). The unsupervised training pipeline is as follows: in each training iteration, every batch of unlabeled images will first go through data augmentation. It can transform the same images into visually different versions. The transformed images will then be fed into two different encoders to get feature arrays. The following Multilayer Perceptron (MLP) will project the feature arrays to embeddings on which the similarity will be calculated. The downstream network adopts a traditional supervised image classification architecture, with a key distinction: the encoder is transferred from the upstream network. This transfer makes the whole network a semi-supervised learning approach, where the network uses both unsupervised knowledge from the upstream network and labeled image data from human. The encoder will firstly extract features from labeled images according to the features it learnt from the upstream network. The MLP structure will then transform the features into high-dimensional embeddings on which the classification loss will be computed. In this

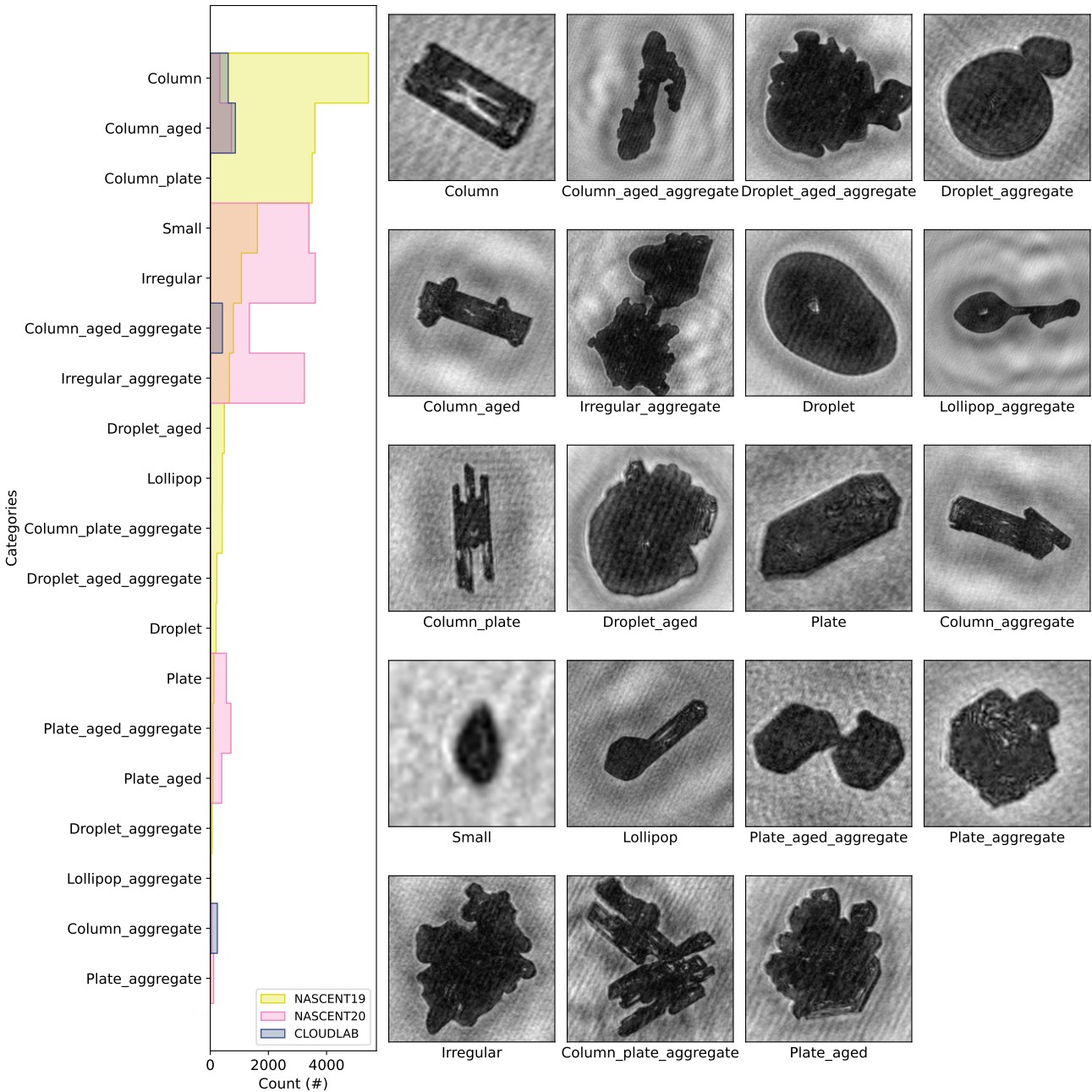

**Figure 1.** The distribution of ice crystal categories in each dataset used in this study is shown in the left most column. The x-axis is the number of images in categories, and the y-axis is the name of category. The NASCENT19, NASCENT20 and CLOUDLAB are indicated by yellow, pink and blue, respectively. For showing the difference in distribution, the order of category is y-axis is sorted in descending order of NASCENT19. The category "column_plate" is the same as "CPC'. The example images of each categories are shown to the right of the distribution map.

section, we will introduce the stages in both networks in the following sections: data augmentation (Sect. 3.1), encoder and projector (Sect. 3.2), loss function (Sect. 3.3), and downstream network (Sect. 3.4).

**Contrastive Semi-Supervised Learning algorithm**

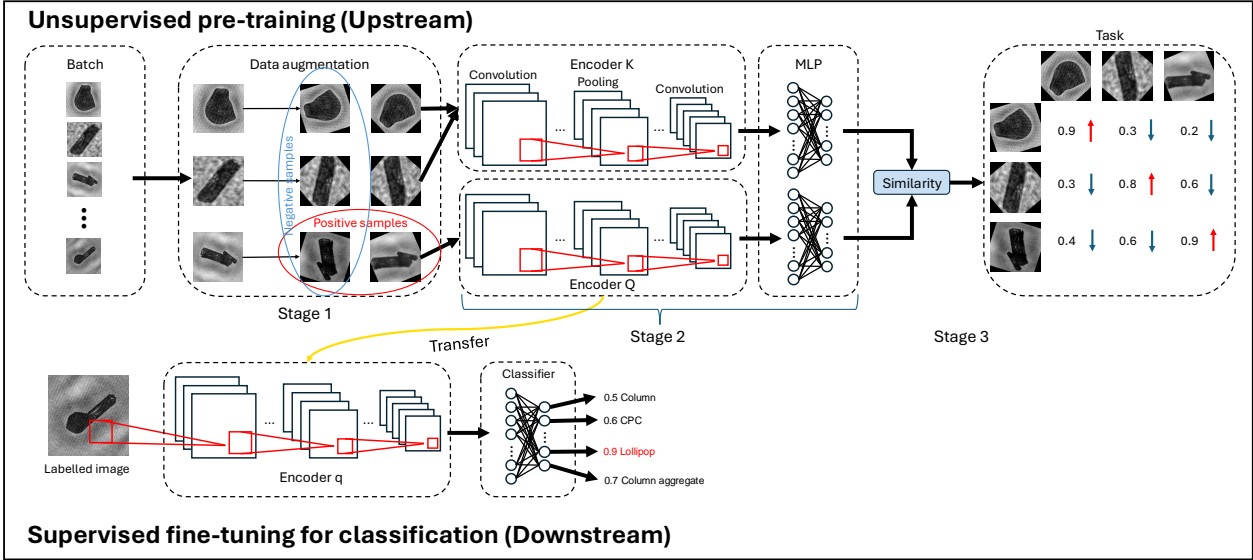

**Figure 2.** The schematic of CSSL algorithm. The dashed squares with the text indicates different stages of both networks. The black arrows show the direction of data propagation in the network. The red squares and lines are the convolution operations in the encoders. The black circles and connect lines in the MLP block indicates the neurons and fully connection among them in different layers. The blue and red oval in the upstream network indicates how to identify positive and negative samples, respectively. It corresponds to the small blue and red arrows in the task block, which is explained as decreasing the similarity between negative samples and increasing the similarity between positive samples.

## 3.1 Data augmentation

Data augmentation can add extra samples to the dataset by converting images to different versions. It is important for unsupervised algorithms (He et al., 2020; Grill et al., 2020; Chen et al., 2020; Chen and He, 2021). It converts original images into visually different images by combining different transformations including random cropping, color jitting, Gaussian blurring,
solarizing, random flipping and random rotation. These transformations are essential to contrastive unsupervised pre-training since they can increase the diversity and quantity original dataset and enhance learning capacity of algorithm (Mumuni and Mumuni, 2022). The examples of applying different transformations are shown in Fig. 3. The input images are uniformly resized to $256 \times 256$ to ensure the consistency during training. Then the images are cropped from edges with a random area ranges from 60 % to 100 % of the input image size and resize it to $224 \times 224$. The lower limit of the cropping area is set
to 60 % because a smaller value will remove the components of an aggregate crystal. For example, the "column_aggregate"

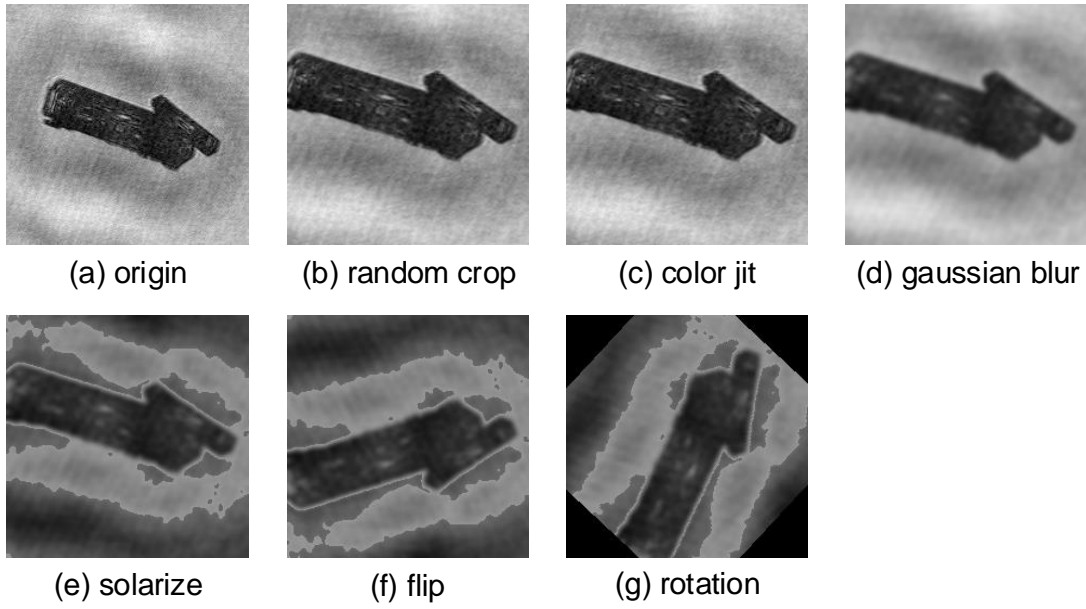

(a) origin    (b) random crop    (c) color jit    (d) gaussian blur

(e) solarize    (f) flip    (g) rotation

**Figure 3.** Examples of data augmentations applied in the upstream network. The subtitle of each picture indicates the transformation superimposed on the previous picture.

becomes a "column"with crop area being lower than 60 %, which change the category of the original image and leads to a false augmentation. After cropping, the image is transformed by a color jitter which randomly change the brightness, contrast, saturation of the image. The subsequent augmentation blurs the image with Gaussian kernel to reduce the level of details in the image. The following augmentation is solarization which inverts the pixel values. Then we randomly flip the image hori-

zontally or vertically, followed by random rotation of the image ranging from 0 to 180 degrees. These manipulations provide different perspectives of the same image. In one batch of training, each image will go though the data augmentation twice, the transformations from the same image are defined as positive samples while the transformations from different images are defined as negative samples, as circled in Fig. 2. By implementing data augmentation, we can generate more different images for the algorithm to compare, which is an effective way for the algorithm to learn useful features for downstream tasks (Geiss

et al., 2024). In the downstream network's supervised fine-tuning, we also employ data augmentation, though its purpose differs from that in contrastive learning. During fine-tuning, data augmentation aims at expanding the size of our training dataset, which prevents overfitting (Shorten and Khoshgoftaar, 2019). The data augmentations during supervised fine-tuning include random cropping and random flipping. The input images in the downstream network are firstly resized to $256 \times 256$, and then they are cropped with area ranging from 60 % to 100 % randomly. Finally, they are randomly flipped before being imputed

into the CNN.

## 3.2 Encoder and projector

The transformed images are then fed into stage 2 where the CNNs first extract their features and then the MLP structure will project the features to embeddings in the latent space. The ability to extract features is facilitated by different layers. Stage 2 in Fig. 2 shows an example of a convolutional neural network. It consists of convolution and pooling layers. After multiple rounds of convolution and pooling, the feature is converted to neurons ($1 \times 1$ size) through an averaging pooling layer. These neurons are then fully connected to the projector, which is a 3-layer MLP (only the input and output layers are shown in Fig. 2). The hidden layer in the middle has 4096 neurons, and the output layer has the dimension of 256 which is also the size of the final embeddings in the latent space.

One of the most representative structures of CNN, the deep residual learning network (ResNet; He et al., 2016) is applied. It solves the problem of gradient vanishing in ultra-deep CNNs (Bengio et al., 1994). The idea of a skip connection between the input and the convolution output ensures the accuracy of the network classification while guaranteeing the depth of CNN. According to different versions of ResNet, the layers within the convolution block and the number of convolution blocks (i.e., the depth of the whole network) will change. There are several versions of ResNet with different numbers of convolution layers. ResNet50 contains 49 convolution layers and 1 fully connected layer, which was proved more efficient and effective than other variations of ResNet (He et al., 2016), and used in this study. A deep network can extract a hierarchy of features, from basic edges and textures in the shallow layers to complex shape patterns in deeper layers (Zeiler and Fergus, 2014). In our task of learning the features of ice crystals, it is necessary to have a sufficiently deep network to extract the detailed information such as the complicated structures of aged particles or aggregates. There is only one difference between the default ResNet50: we changed the output dimension of the output layer to 256 (as mentioned before) rather than 2048 used in the original ResNet50.

The upstream network has two parallel encoders: online encoder $Q$ and target encoder $K$. When the network forward propagates the input, one of the positive samples will be encoded by $Q$, the other positive sample, and the rest of the negative samples will be encoded by $K$. The design of parallel encoders aims to teach the network to distinguish between positive and negative samples, which is the most common design in contrastive learning.

## 3.3 Similarity functions

The similarity function in the upstream network serves as the loss function between the inputs. We evaluated two typical unsupervised contrastive learning structures in this study: MoCo (He et al., 2020) and BYOL (Grill et al., 2020). The main differences between the two models are: 1. MoCo has a "memory bank" to store embeddings of previous batches obtained from target projector so that the embeddings from online encoders have sufficient samples to compare with while BYOL does not have the "memory bank". 2. The way to calculate the similarity is different. MoCo computes the similarity using Information Noise-Contrastive Estimation (InfoNCE) loss function (Oord et al., 2018):

$$L_{q,k} = -\sum_{j=0}^{B+1} \log \frac{\exp(q \cdot k_0 / \tau)}{\sum_{i=1}^{B+1} \exp(q \cdot k_i / \tau)} \cdot y_j \tag{1}$$

where $q$ and $k_0$ represent the embeddings of positive samples that go through online encoder $Q$ and target encoder $K$, respectively, and $k_i$ is the embeddings of negative samples. $y_j$ is the one-hot vector containing 1 positive indicator (number 1) and $B$ negative indicators (number 0) because in each batch there is only one positive sample and the rest are negative samples, and $\tau$ is known as the "temperature" parameter, which scales the value of $q \cdot k_i$, making it easier to optimize the network. BYOL uses cosine similarity:

$$L_{q,k} = 2 - 2 \cdot \frac{\langle q, k \rangle}{\|q\|_2 \cdot \|k\|_2}. \tag{2}$$

the $q$ and $k$ are the embeddings of positive samples and negative samples in each batch.

After the similarity is calculated, the similarity gradient will propagate backward and will be used to update the weights of the encoders and projectors. Noticeably, the gradient will propagate only through the online encoder and projector. The target encoder $K$ do a momentum update of the weight from the online encoder $Q$:

$$\theta_K \leftarrow \tau \theta_K + (1 - \tau)\theta_Q \tag{3}$$

where $\theta$ is the weight matrix, the suffix $Q$ and $K$ indicates the encoders and the parameter $\tau$ defines the momentum, namely the degree of change from the weight of the online encoder. The introduction of momentum update is to solve the training failure caused by the rapidly changing encoder that reduces the consistency of embeddings in the target encoder $K$ (He et al., 2020).

## 3.4 Downstream network

The downstream network is where we train the algorithm for the classification task. We transfer the trained encoder $Q$ from the upstream network, which theoretically has the preliminary ability to classify different images after training in the upstream network. The transferred encoder $Q$ is connected to a classifier which is a 3-layer MLP. It is different from the MLP in the upstream network that the size of the output layer is equal to the number of categories. The outputs will be activated by Softmax function that maps the outputs to probabilities ranging from 0 to 1. The category with maximum probability will be the prediction of the network. And then the cross-entropy loss is calculated between predictions and true labels:

$$L_x = -\sum_{n=1}^{N} \sum_{c=1}^{C} y_{n,c} \log x_{n,c} \tag{4}$$

where $N$ is the size of a batch, $C$ is the number of categories, $x_{n,c}$ is the probability that input $n$ is of category $c$, and $y_{n,c}$ is the binary indicator 0 (1) if category $c$ is incorrect (correct) for input $n$. The weights of the network are updated by the back-propagation of the loss function. One major difference from the upstream network is that the input images in the downstream network are labeled. The labeled images not only act as the ground truth when we fine-tune the classification network but also are the physical knowledge of the ice crystals that we input to the encoder $Q$ to generate more accurate predictions. The involvement of part of human knowledge (i.e. image labels) in the downstream network is the reason why we recognized the algorithm is semi-supervised.

### 3.5 Evaluation

The evaluation of our algorithm will primarily focus on the performance of the downstream network, with a quantitative emphasis on the classification accuracy across different models. Additionally, we will qualitatively analyze the upstream network by visualizing high-dimensional embeddings in a 2D space using the t-distributed stochastic neighbor embedding (t-SNE) technique. It is a dimension reduction method that aims to retain as much of the important structure of the high-dimensional data as possible in the low-dimensional representation.

#### 3.5.1 Cross-validation

A 5-fold cross-validation was used when training the downstream network for classification. Cross-validation reduces uncertainty in model performance while retaining as much data as possible for training. In a 5-fold cross-validation, the dataset used for training is randomly re-sampled into 5 subsets (folds). In each round of training, 4 of the 5 folds will be used for training, and the remaining folds will be used for validation. This process will be repeated 5 times, and each time a different fold will be used for validation.

#### 3.5.2 Generalization ability

One important aspect of a model that should be evaluated is its ability to generalize the knowledge learned in the previous data set to a new dataset. This ability to reuse knowledge is known as generalization ability in deep learning (Jiang et al., 2022). In this study, the generalization ability will be tested on new dataset that has not been used for training. For example, the case in our study would be the downstream network trained on NASCENT19 is tested on CLOUDLAB.

#### 3.5.3 Metrics

There are several metrics to evaluate the classification performance. One of them is overall accuracy (OA), which is the percentage of correctly classified images over the total number of images:

$$OA = 100\% \cdot \frac{1}{N} \sum_{i=1}^{N} l\{\hat{y}_i = y_i\} \tag{5}$$

$N$ is the total number of images in a dataset. $\hat{y}_i$ is the prediction of downstream network, while $y_i$ is the true label of the input image. $l$ is an indicator function. It equals to 1 if $\hat{y}_i = y_i$ and 0 otherwise. OA is 100 % if all images are correctly classified.

Precision and recall measure the accuracy of the model on each category. The equations of per-category precision and recall are:

$$precision_i = \frac{TP_i}{TP_i + FP_i} \tag{6}$$

$$recall_i = \frac{TP_i}{TP_i + FN_i} \tag{7}$$

The subscript $i$ represents one specific category. The number of images that are correctly predicted as $i$ is the true positives ($TP_i$) for category $i$. The number of images that are predicted as $i$ have different labels are false positives ($FP_i$) for category $i$. The number of images in category $i$ that are not correctly predicted as $i$ are false negatives ($FN_i$). The per-category precision $precision_i$ represents the ratio of correctly predicted images to all images that are predicted as $i$. The per-category recall $recall_i$ represents the ratio of correctly predicted images to images with true label $i$.

A high per-category precision indicates that the model is good at predicting a specific category, while a high per-category recall indicates the model is good at identifying images from a specific category. F1-score is the harmonic mean of precision and recall, which reflect the combine effect of precision and recall:

$$F1_i = 2 \cdot \frac{precision_i \cdot recall_i}{precision_i + recall_i} \tag{8}$$

The OA can be also defined as:

$$OA = 100\% \cdot \frac{1}{N} \sum_{i=1}^{N} TP_i \tag{9}$$

The metrics mentioned above can be unified and visualized in one diagram: confusion matrix. As shown in Table 3, each row represents a label and each column represents a prediction. The diagonal are the TPs of each category. Expect for the diagonal, the sum of each row is the number of FNs, and the sum

| | **Prediction 1** | **Prediction 2** | **Prediction 3** | **Prediction 4** | **Prediction 5** |
|---|---|---|---|---|---|
| **Label 1** | $TP_1$ | – | – | – | – |
| **Label 2** | $FN_2/FP_1$ | $TP_2$ | – | – | – |
| **Label 3** | $FN_3/FP_1$ | $FN_3/FP_2$ | $TP_3$ | – | – |
| **Label 4** | $FN_4/FP_1$ | $FN_4/FP_2$ | $FN_4/FP_3$ | $TP_4$ | – |
| **Label 5** | $FN_5/FP_1$ | $FN_5/FP_2$ | $FN_5/FP_3$ | $FN_5/FP_4$ | $TP_5$ |

**Table 3.** An example of confusion matrix

## 4 Implementation

### 4.1 Training configurations

Since we have two different parts of the network in our algorithm, they follow different training configurations. Since the upstream network of the CSSL algorithm usually requires large batch sizes ($\geq 256$), we ran our algorithm on a high-performance cluster with 4 RTX3090 GPUs, with a batch size of 64 on each GPU (in total 256), which reach the maximum memory usage of one GPU. Beyond the GPU requirements, the algorithm requires a minimum computing environment consisting of a 4-core

CPU and 16GB of system memory to operate.To cope with the large dataset inputted into encoders, we applied the LARS (Layer-wise Adaptive Rate Scaling) optimizer to ensure stable and efficient training (You et al., 2017). When we fine-tuned the downstream network, we applied the standard stochastic gradient descent optimizer on the same GPUs with 64 batch sizes on each of them.

In terms of learning rates, instead of using one constant learning rate throughout the training process, we design a scheduler that changes the learning rate during training. The specific strategy is that the learning rate will increase linearly from a very small value ($1 \times 10^{-5}$) to a target value (e.g. $0.01$) in the first several iterations, and then the learning rate decrease gradually to $0$ according the cosine annealing:

$$lr_t = \frac{1}{2} \cdot lr_{t-1} \cdot (1 + \frac{t \cos \pi}{T}) \tag{10}$$

where $t$ is the current iteration, $T$ is the number of total iterations. Both networks follow the same learning rate scheduler.

### 4.2 Experiments

#### 4.2.1 The effect of the training set size

In the upstream network, we performed unsupervised pre-training using unlabeled NASCENT19 and NASCENT20 (in total 33354 images) to obtain models for transferring to the downstream network. In the upstream network, we use two structures: the MoCo and the BYOL, as described in Sect. 3. These two models are named Unsup-MoCo and Unsup-BYOL, respectively. In the rest of paper, we will refer to a model by its type: unsupervised (unsup), supervised (sup) and semi-supervised (semisup); and its specific structure: MoCo and BYOL. The 'unsupervised' here represents the upstream network of CSSL algorithm. The 'semi-supervised' specifically refers to the downstream network of CSSL. 'Supervised" refers to purely supervised models. Before further pre-training on the NASCENT datasets, the weights of both models are initialized with the weights of the respective structures pre-trained on the imagenet-1k dataset: IM1K-Unsup-MoCo and IM1K-Unsup-BYOL. In this paper, the weight initialization will be refereed like 'dataset-type-structure'. In the downstream network (supervised classification) stage, we conducted several experiments using the NASCENT19 dataset with training sets of different sizes (Table 4). The weights of the downstream networks of the two models were initialized by the weights of the online encoders pre-trained on the NASCENT dataset in their respective upstream networks: NASCENT-Unsup-MoCo and NASCENT-Unsup-BYOL, which represent the transfer of the online encoder to the downstream network. We also included the performance of purely supervised models and IceDetectNet for comparison. The former acts as the baseline model for this study, and the later one is the latest supervised algorithm for ice crystal classification. For fair comparison, the baseline models used the same weight initialization as Unsup-MoCo: IM1K-Unsup-MoCo. To make it easier to refer to the models, we have given each model a short name based on its structure, the size of training set (n), and the number of categories (c) in the training set: $[\text{structure}]^n_{\text{sup/unsup/semisup},c}$. For example, the Unsup-MoCo has a short name: $[\text{MoCo}]^{33354}_{\text{unsup},19}$. The Semisup-MoCo trained on 128 images and 19 categories is: $[\text{MoCo}]^{128}_{\text{semisup},19}$. The evaluation of different models will follow the metrics mentioned Section in 3.5.3. Firstly, under each size of training set, the accuracy of each individual model in the 5-fold cross-validation will be calculated. Secondly, the per-category accuracy will be computed and demonstrated as confusion matrix.

| Name | Dataset | Weight initialization | Size of training set (n) | Categories (c) |
|---|---|---|---|---|
| Semisup-MoCo | NASCENT19 | NASCENT-Unsup-MoCo | [128, 256, 512, 1024, 2048, 4096, 8192, 16384, 18864] | 19 |
| Semisup-BYOL | NASCENT19 | NASCENT-Unsup-BYOL | [128, 256, 512, 1024, 2048, 4096, 8192, 16384, 18864] | 19 |
| Sup (Baseline) | NASCENT19 | IM1K-Unsup-MoCo | [128, 256, 512, 1024, 2048, 4096, 8192, 16384, 18864] | 19 |
| IceDetectNet | NASCENT19 | IM1K-Sup | 18864 | 19 |

**Table 4.** The models trained and used for studying the effect of training set size. "Semisup" represent the classification stage of CSSL which used both knowledge from unsupervised pre-training and image labels. "Sup" means that the models are purely supervised. "MoCo" and "BYOL" represent the encoders transferred from "Unsup-MoCo" or "Unsup-BYOL". IceDetectNet is the latest supervised algorithm from (Zhang et al., 2024).

| Name | Dataset | Weight initialization | Size of training set (n) | Categories (c) |
|---|---|---|---|---|
| Semisup-MoCo-4CAT | NASCENT19-4CAT | NASCENT-Unsup-MoCo | [128, 256, 512, 1024, 2048, 4096, 8192] | 4 |
| Semisup-BYOL-4CAT | NASCENT19-4CAT | NASCENT-Unsup-MoCo | [128, 256, 512, 1024, 2048, 4096, 8192] | 4 |
| Sup-4CAT (Baseline-4CAT) | NASCENT19-4CAT | IM1K-Unsup-MoCo | [128, 256, 512, 1024, 2048, 4096, 8192] | 4 |

**Table 5.** The same as Table 4, except that the downstream network and baseline model were trained using NASCENT19 dataset with 4 categories.

#### 4.2.2    The effect of the number of categories

To explore the effect of the number of categories on the classification performance of models, we extract 4 categories from NASCENT19: 'Column', 'Column_aged', 'Column_aggregate' and 'Column_aged_aggregate', which are the same as the categories in CLOUDLAB dataset. The new dataset NASCENT19-4CAT contains 9,182 images. In addition to the models listed in Table 4, we trained additional models on NASCENT19 with 4 categories. The extra models trained are shown in Table 5. The weight initialization for Semisup-MoCo-4CAT is identical to that of Semisup-MoCo. Similarly, Semisup-BYOL share the same weight initialization as Semisup-BYOL. The weight initialization for Baseline-4CAT is also consistent with previous baseline models.

We began by comparing the cross-validation results of the same models on both the 19-category and 4-category datasets, which will provide us with the general impact of varying the number of categories. Then we will compare across the performance of different models in the 4-category classification task, in order to discover whether the difference between models would be affect by the number of categories.

#### 4.2.3    Evaluate the generalization ability

The generalization ability of CSSL algorithm will be evaluated by testing the models trained with the 4-category dataset NASCENT19-4CAT on a new dataset: CLOUDLAB. Since the CSSL and baseline models were trained with different sizes of the 4-category dataset, and each case has 5 models due to cross-validation, each single model will be tested with the entire CLOUDLAB dataset to obtain the accuracy. Therefore, the results will be displayed in the same format as cross-validation results.

## 5    Results

### 5.1    Performance of classification models on different training set sizes

The performance of classification is first evaluated by the overall accuracy using the 5-folds cross-validation mentioned in Sect. 3.5.1. The effect of the size of the training set on overall accuracy is shown in Fig. 4. Among all models, the OA increases with the size of training set as expected. In general, the Semisup-MoCo models exhibit higher OA compared to the baseline models, while the Semisup-BYOL models have lower OA than the baseline models. It shows that the MoCo structure performs better on the task of classifying 19 categories of ice crystal images, and the following analysis will focus between MoCo models and baseline models. As shown in Fig. 4, the OA difference between the Semisup-MoCo models and the baseline model is larger when the training set is small ($n \leq 2048$). When the size of the training set exceeds this threshold ($n > 2048$), the differences of OA between the Semisup-MoCo models and the baseline models narrow. This suggests that the CSSL algorithm performs better when the number of labeled ice crystal images available for training is limited. The accuracy range of both the baseline and CSSL models trained on small datasets ($n < 2048$) is higher than that trained on large datasets ($n > 2048$), which shows that the models are unstable when they are trained with a small dataset size. One possible reason we concluded from

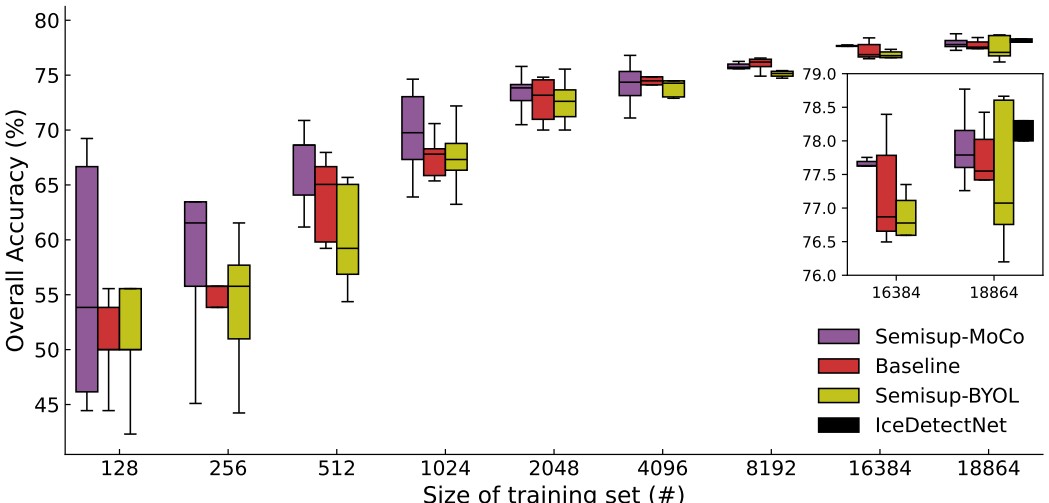

**Figure 4.** Dependency of the overall accuracy of various models on the training set sizes. The central black lines inside the boxes are the median values of accuracy of each 5-fold cross-validation experiment on the training set size. The lower limit and upper limit of boxes are the first quartile and the third quartile, respectively. The range of boxes shows the distribution of central 50 % accuracy values, which represents the average performance of each model. The error bars show the maximum and minimum accuracy values from each 5-fold cross-validation experiment. The inset figure zooms in on the results of 16384 and 18864 samples. The IceDetectNet from (Zhang et al., 2024) (green box) is shown only for the training set size of 18,864 samples.

checking the loss tendency during supervised fine-tuning on different sizes of dataset (Fig. A2) is that the models fine-tuned on small sizes of dataset ($n < 2048$) is suboptimal compared to models fine-tuned on larger sizes, which would lead to unstable classification performance. Another possible reason we concluded from the loss value of unsupervised pre-training (Fig. A1) is that the 33354 images may not be sufficient for optimizing the upstream network, which means the classification performance of CSSL algorithm could be further improved even when fine-tuning small size dataset if we pre-trained with more ice crystal images. We include the loss values in Appendix A. The $[\text{MoCo}]_{\text{semisup},19}^{18864}$ reaches a comparable OA ($77.9\% \pm 0.58\%$) as the IceDetectNet ($78.2\% \pm 0.9\%$). This indicates that the performance of the CSSL algorithm is comparable to IceDetectNet, which is currently the top model on the NASCENT19 dataset.

It is time-consuming to manually label ice crystal images for training models, especially when deciding on labels from many categories. According to authors' experiences of labeling ice crystal images, it took 6 seconds in average to manually label images of basic habits, while it took 60 seconds in average for the images of complex shapes. We assumed an average value of 33 seconds for manually labeling one image. If we take the OA of $[\text{Baseline}]_{\text{sup},19}^{18864}$ as a reference OA (ROA), we define a decreased overall accuracy (DOA) as the difference between the OA of each Semisup-MoCo model ($[MoCoOA]$) and the reference OA:

$$DOA_n = ROA - [MoCoOA]_n \tag{11}$$

where $n$ is the size of training set. We also defined the time spent labeling 18864 images as a reference time (RT) and calculated the difference between the time spent labeling other sizes of training sets and the reference time as the time saved (TS) for each training set:

$$TS_n = RT - T_n \tag{12}$$

The results are shown in Fig. 5. The more samples in the training set, the smaller the DOA and the less time saved on manual labeling. We find an inflection point, which represents $[\text{MoCo}]_{\text{semisup},19}^{2048}$, which saves 154 hours of manual labeling time at the expense of just 3.8 % accuracy

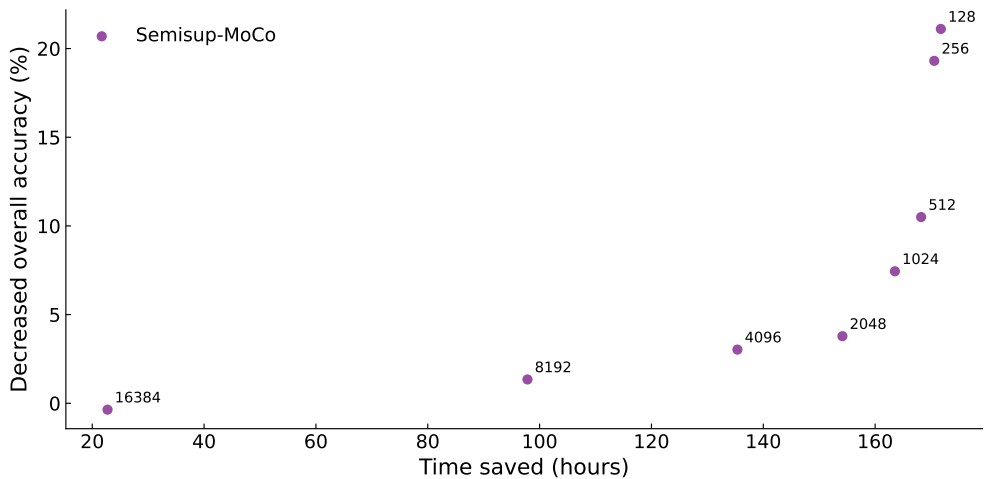

**Figure 5.** The figure illustrates the relationship between DOA (y-axis) and time saved for manual labeling (x-axis). Each dot represents a cross-validation of a Semisup-MoCo model with a different training set size (marked on the left of each data point). A higher time saved indicates a smaller training set and a correspondingly higher DOA, whereas a lower time saved suggests a larger training set and a lower DOA.

We further analyzed the performance in terms of precision and recall of basic habits and complex shapes separately in a confusion matrix. The confusion matrix was calculated on the validation set for each fold and then averaged over all 5 folds so that there will be one confusion matrix per experiment. We used the results of $[\text{MoCo}]_{\text{semisup},19}^{18864}$ and $[\text{Baseline}]_{\text{sup},19}^{18864}$ as examples. Figure 6 shows the confusion matrix of MoCo models (left) and baseline models (right) on basic habits. The overall accuracy of $[\text{MoCo}]_{\text{semisup},19}^{18864}$ on basic habits (94.11 %) is close to $[\text{Baseline}]_{\text{sup},19}^{18864}$ (94.04 %). Both models have highest precision and recall rates ($> 90\%$) on basic habits such as "Column", "small", "Column_plate (CPC)" and "Droplet", which means they are the main contributors of overall accuracy. Meanwhile, these categories have the largest share in NASCENT19 except for "Droplet".

Figure 7 shows the confusion matrix of both models in complex shapes. In general, the precision and recall rates for complex shapes in $[\text{Baseline}]_{\text{sup},19}^{18864}$ are lower than those in $[\text{MoCo}]_{\text{sup},19}^{18864}$, and the overall accuracy follows the same trend. However, the

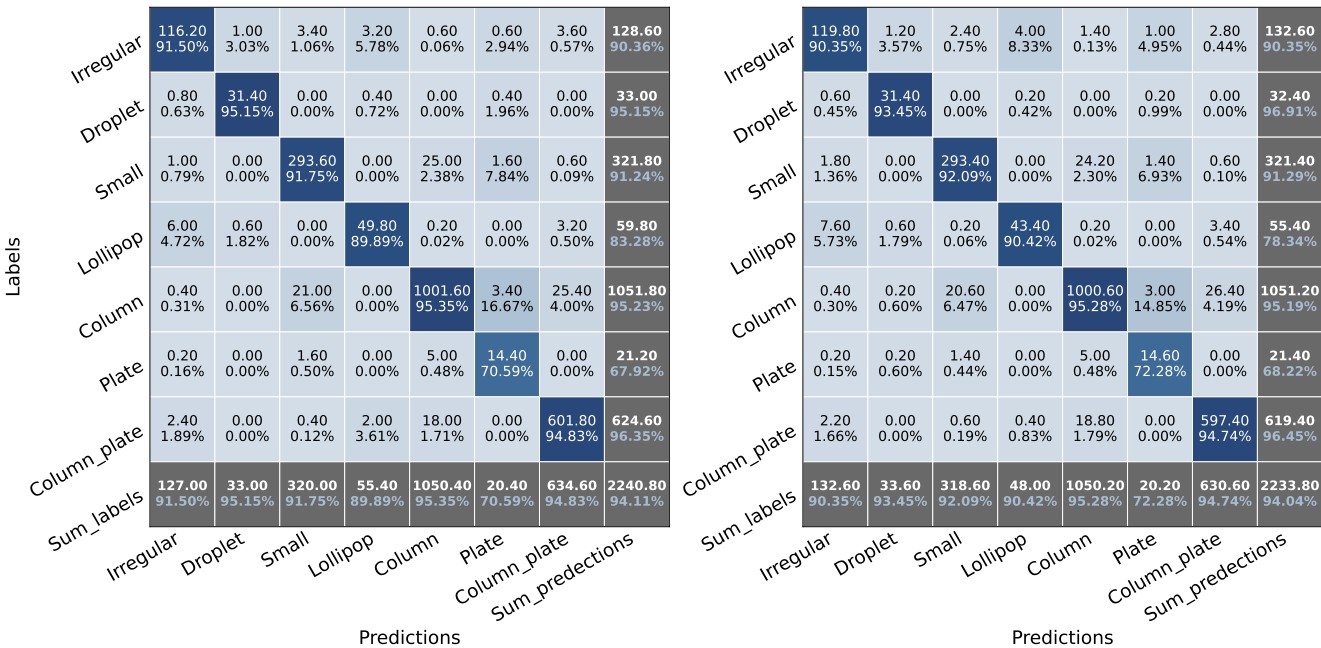

**Figure 6.** The confusion matrix of $[\text{MoCo}]_{\text{semisup},19}^{18864}$ (left) and $[\text{Baseline}]_{\text{sup},19}^{18864}$ (right). The two confusion matrix are calculated based on the six basic habits. The text on the left side of each confusion matrix shows the actual labels of categories, and the text on the bottom is for the predicted categories. As shown in Fig. 3, the diagonal are the true positives of each category. The numbers and percentages in the last row are the sum of predicted categories (in white) and precision of each category (in blue), while the numbers and percentages in the last column are the sum of actual categories (in white) and recall of each category (in blue). The percentage (in blue) in the right bottom corner represents the overall accuracy. The percentages in the rest of grids are the ratio of predicted samples to the sum of predict labels of each corresponding column.

performance of $[\text{MoCo}]_{\text{sup},19}^{18864}$ and $[\text{Baseline}]_{\text{sup},19}^{18864}$ in complex shapes is not as good as it is in basic habits. The precision and recall rates on "aged" categories ($\sim 90\%$) are much higher than the "aggregate" and "aged_aggregate" (65 % to 60 %). This indicates that models are more capable of predicting aged ice crystals. If we calculated the percentage of these 3 types of ice crystals in the NASECENT19, we can find that "aged" ice crystals accounts for 22.1 %, which follows after the percentage of "column": 28.8 %. However, "aggregate" and "aged_aggregate" only accounts for 6.34 % and 5.91 %, respectively. The

categories with high precision and recall rates have one thing in common: they all have a higher share of the number in NASCENT19 than the other categories.

  We also analyzed the confusion matrices of models trained on datasets of different sizes (not shown). The main conclusions are similar. First, in general, the two models have similar overall accuracy of basic habits. Second, the Semisup-MoCo models perform better than the baseline models for complex shapes. Third, all models have higher precision and recall rate on the

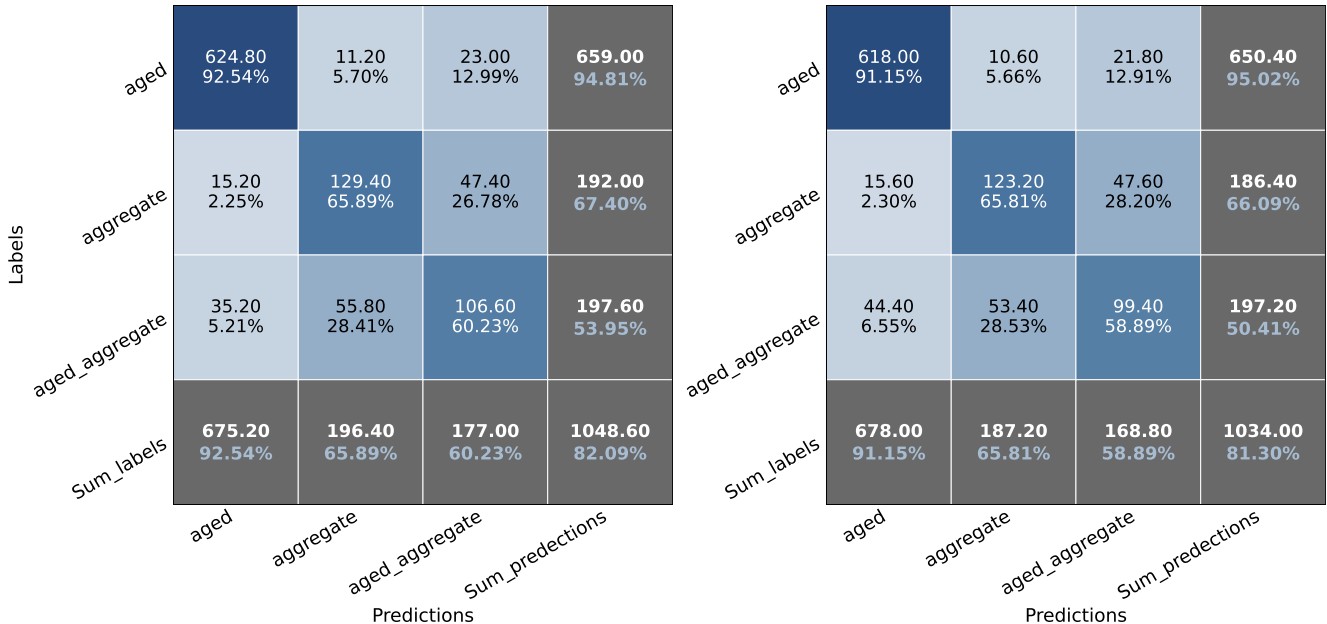

**Figure 7.** The same as Fig. 6 but the confusion matrix is calculated on the complex shapes of ice crystals. The rest 13 complex shapes are merged down to 3 types here: "aged", "aggregate", "aged_aggregate", which represents the microphysical processes happened to the ice crystals.

categories that dominate the dataset. The advantages of the CSSL algorithm in predicting complex shapes originates from the fact that the encoder learns useful representations from the unsupervised pretraining.

Figure 8 presents 2-dimensional visualizations of the embeddings generated by the encoder of the upstream network with a MoCo structure. These visualizations are shown at three stages: before unsupervised pre-training (a-d), after unsupervised pre-training (e-h), and after supervised fine-tuning (i-l), using the t-SNE technique. Each column of visualizations represents the results for different training sizes, arranged from left to right as follows: 18864, 8192, 4096, and 2048 samples. It can be found before any training (a to d) that there are no clusters formed for any category as expected, as the dots with different colors are mixed up together. Some clusters have formed after the unsupervised pre-training, which shows the ability of the upstream network to distinguish between some categories. However, the boundaries between the different categories are not clear, in other words, the inter-category distance is small, which reveals the fact that some images are difficult to classify into one particular category because they have visually similar features. The figures in the third row present the 2D visualizations generated by the Semisup-MoCo models. After fine-tuning these models with labeled data, they demonstrate a significantly improved ability to distinguish between categories that were previously mixed in the 2D visualization maps produced by the Unsup-MoCo models. For instance, categories such as "Column", "Column_aged", and "Column_aged_aggregate" which were difficult to separate earlier, now form three distinct clusters. However, some complex shapes remain intermixed. In addition to

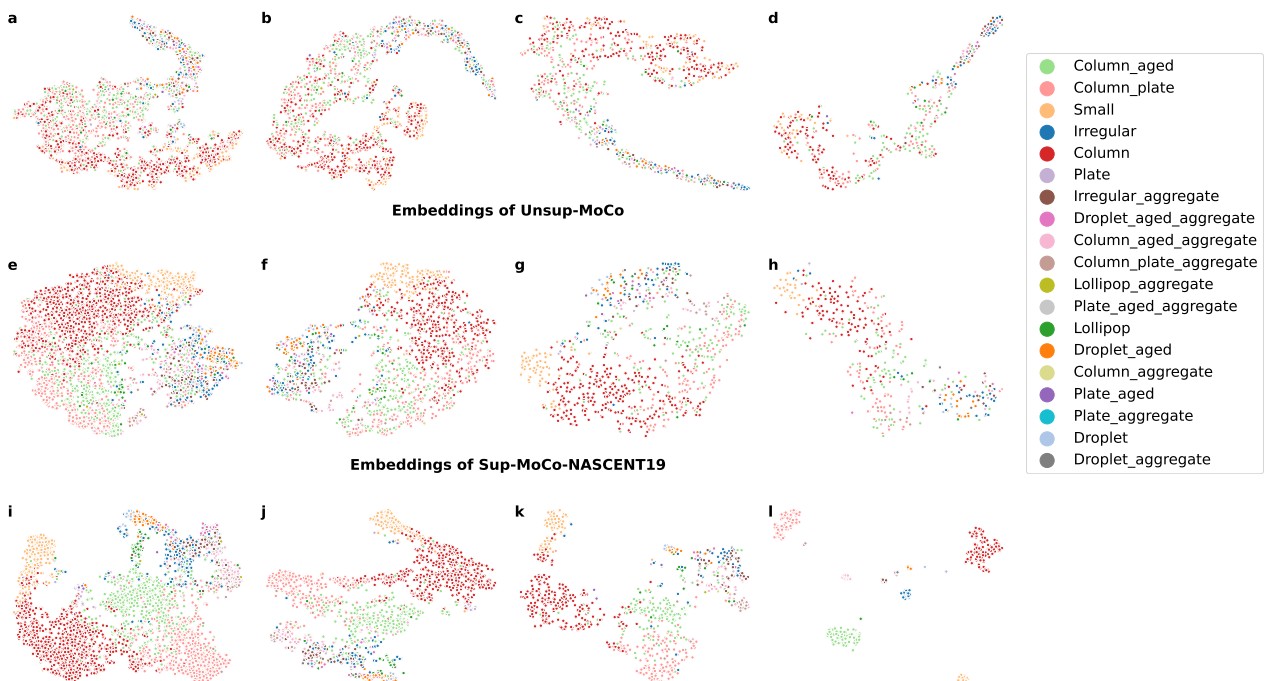

**Figure 8.** The t-SNE plots visualize the embeddings from the encoder of upstream network. Figures a to l display the embeddings of the validation set from the first fold of each experiment, corresponding to different sizes of training sets of different models. The first row (a to d) shows the t-SNE plots of the data embeddings before any training, and the training set sizes are 18864, 8192, 4096, and 2048, respectively. The second row (e to h) presents the t-SNE plots of embeddings of Unsup-MoCo, and the training set sizes from e to h are the same as a to d. The third row (i to l) is the t-SNE plots of embeddings of Semisup-MoCo models.

the primary clusters in red ("Column"), green ("Column_aged"), pink ("Column_aged_aggregate"), and yellow ("small"), we identified an additional cluster that contains a mix of different complex shapes.

## 5.2 Performance of classification model on different number of categories

Figure 9 shows the performance of models on NASCENT19-4CAT. The results closely resemble those on NASCENT19, with the key difference being higher overall accuracy compared to the results of the 19-category classification. In general, both Semisup-MoCo-4CAT and Semisup-BYOL-4CAT have higher OA than the baseline models, indicating that the CSSL algorithm is better than the baseline models in a 4-category classification task. The overall accuracy gap between CSSL models and the baseline models narrows when the size of the training set exceeds 512. The overall accuracy of $[\text{MoCo}]^{2048}_{\text{semisup},4}$ (89.6 %) is comparable to $[\text{Baseline}]^{8192}_{\text{sup},4}$ (90.1 %) considering the standard deviation. We demonstrated that using only 25 % of the NASCENT19-4CAT images (2048) in the CSSL algorithm achieves performance comparable to that of a fully supervised

410 model trained on the entire NASCENT19-4CAT dataset. The accuracy range of both the baseline and CSSL models trained on small datasets (n < 2048) are large. The reason is the same as the models fine-tuned on 19-category dataset. The relationship between time saved and decreased overall accuracy is also evaluated for Semisup-MoCo-4CAT, as shown in Fig. 10. The inflection in the case of 4-category classification represents for the $[\text{MoCo}]^{2048}_{\text{semisup},4}$. The model saves 56 hours at the expense of 2.5 % of overall accuracy.

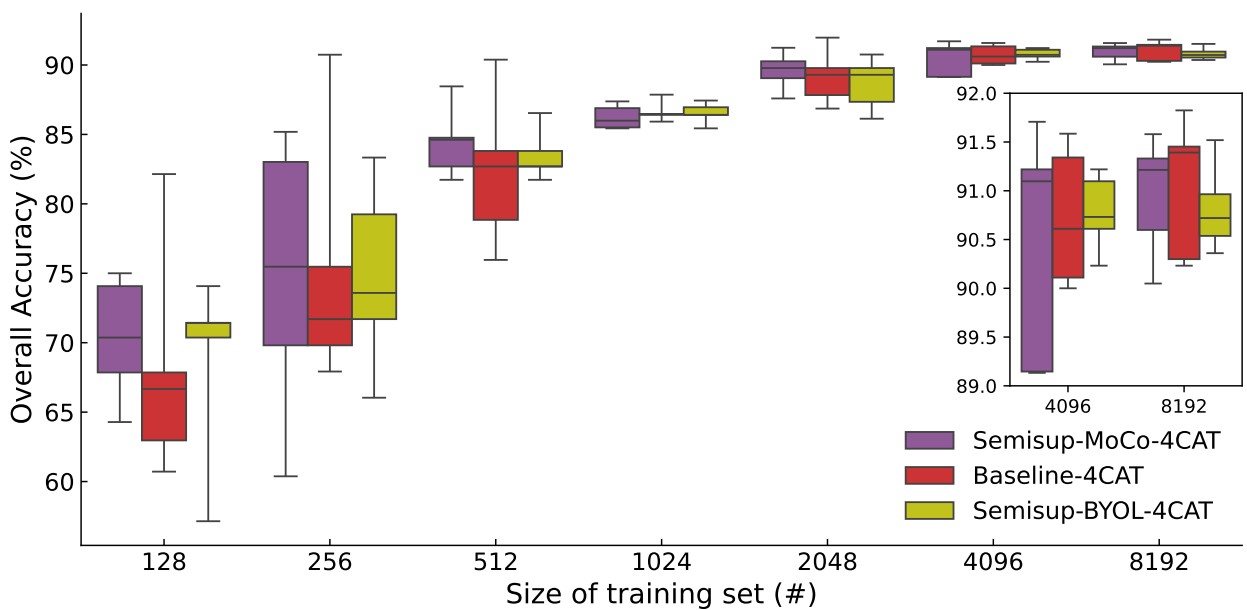

**Figure 9.** The same as Fig. 4 except the results are evaluated on NASCENT19-4CAT. The result from IceDetectNet is not shown here because it was not trained on NASCENT19-4CAT. The maximum size of training set here is set to 8192 instead of 16384 because the size of NAsCENT19-4CAT is smaller than 16384.

415 In terms of per-category performance, we also analyzed the confusion matrix for $[\text{MoCo}]^{8192}_{\text{semisup},4}$ and $[\text{Baseline}]^{8192}_{\text{sup},4}$ in Fig. 11. The overall accuracy of two models are similar. However, $[\text{MoCo}]^{8192}_{\text{semisup},4}$ has a better performance on complex shapes in the 4-category classification task. Both models tends to have higher precision and recall rate on "Column", "Column_aged" and "Column_aged_aggregate" that are the major categories in NASCENT19-4CAT. It indicates that after we reduce the number of categories, both models perform better on the categories that account for a large percentage in the dataset. In order to

420 test the models on a dataset that is not used in both unsupervised pre-training and supervised fine-tuning, we used them to test on CLOUDLAB dataset and evaluate the performance without any further fine-tuning. The results are shown in Fig. 12. The Semisup-MoCo-4CAT models have better performance than both Baseline and Semisup-BYOL-4CAT models, and the Semisup-BYOL-4CAT models are ever worse than Baseline models, which indicates that the models based on BYOL structure are not good at generalizing on unseen dataset in training. The gap of overall accuracy between Semisup-MoCo-4CAT and

425 Baseline models is considerably large when the size of training set is smaller than 1024. It shows a better generalization ability

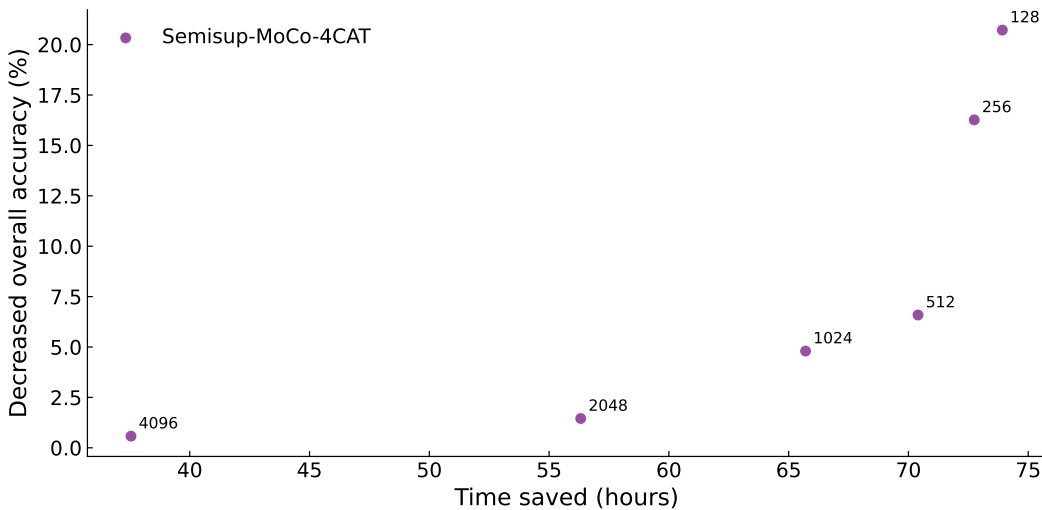

**Figure 10.** The same as Fig. 5 but for Semisup-MoCo-4CAT.

of CSSL algorithm based on MoCo structure than Baseline models when the size of training set is small. When the size is larger than 1024, except for the case of 4096, the difference between Semisup-MoCo-4CAT and Baseline models are within 1 %, which indicate the generalization ability difference between these two models are basically eliminated if they are trained on a larger size of dataset. We also found that the $[\text{MoCo}]_{\text{semisup},4}^{4096}$ has reached the same test overall accuracy on $[\text{Baseline}]_{\text{sup},4}^{8192}$, which again proved the strong generalization ability of CSSL algorithm trained on half of the entire NASCENT19-4CAT.

## 6 Conclusions

We propose a contrastive semi-supervised algorithm for ice crystal classification, employing a two-stage training paradigm. First, the model undergoes unsupervised pre-training to learn general features from unlabeled ice crystals. Then, it is fine-tuned by hand-labeled ice crystals to conduct classification. The algorithm is pre-trained on two datasets, both collected during the NASCENT campaign, and fine-tuned and evaluated on one of them using different dataset sizes. Additionally, it is further assessed on the 4-category NASCENT19 dataset, also with varying sizes. The resulting models are then tested on a new dataset collected during the CLOUDLAB project to evaluate their performance on unseen data. The evaluation on NASCENT19 datasets shows that the overall classification performance of CSSL algorithm is better than the a baseline model which is a purely supervised algorithm. In terms of small sizes of datasets ($\leq 2048$ images), the overall accuracy of CSSL algorithm exceeds baseline models 3 % in average. In terms of 19-category dataset, the performance of CSSL algorithm trained on 2048 images show that the algorithm can save 90 % of time on manual labeling while maintain the drop in overall accuracy within 5 %. In addition, the CSSL algorithm trained on the entire dataset achieved $77.9\%\pm0.58\%$ overall accuracy which is comparable to the IceDetectNet ($78.2\%\pm0.9\%$) which was the best supervised algorithm on the same dataset. In terms of per-category

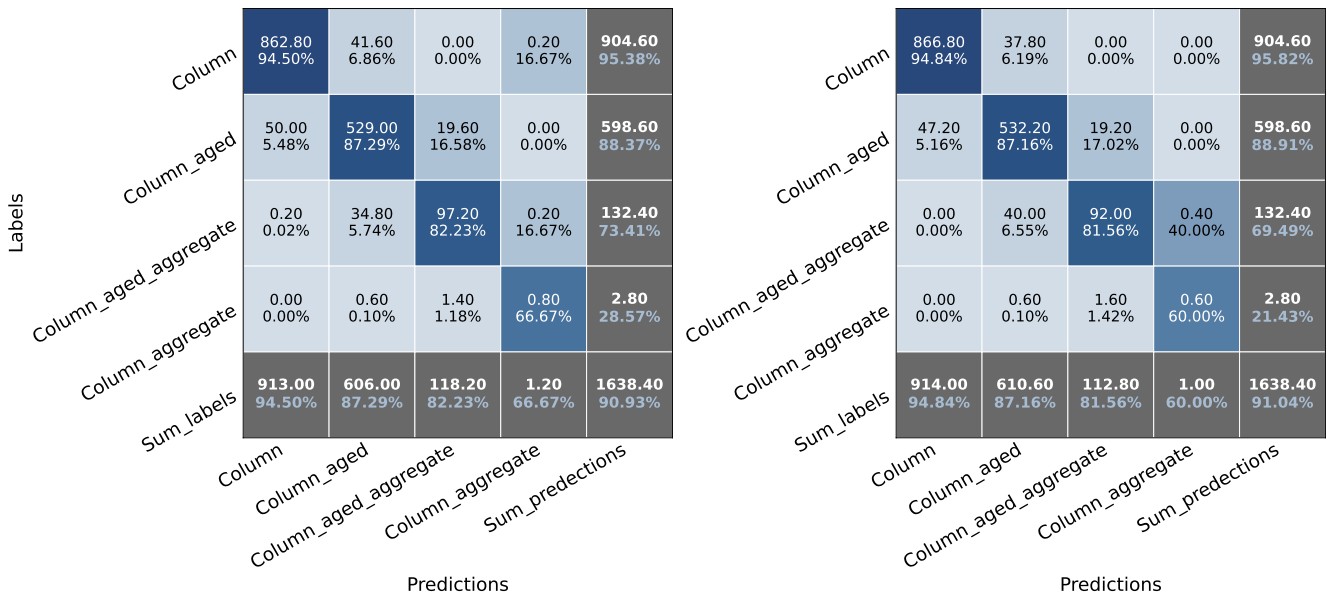

**Figure 11.** The same as Fig. 6 and Fig. 7 but for NASCENT-4CAT. The confusion matrix is not divided into basic habits as complex shapes because there are only 4 categories in the dataset.

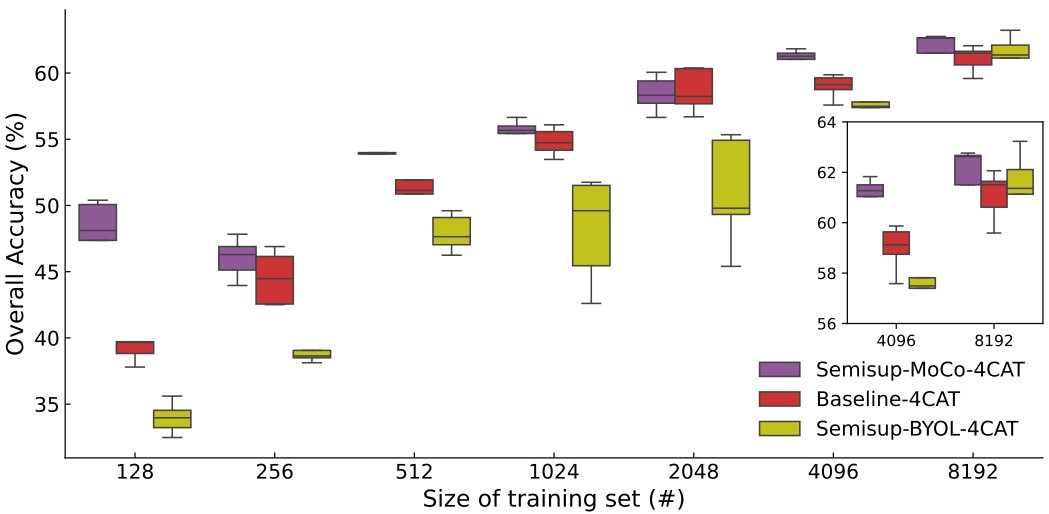

**Figure 12.** The overall accuracy of Semisup-MoCo-4CAT, Semisup-BYOL-4CAT and Baseline models tested on the CLOUDLAB dataset without any further fine-tuning. The CSSL and Baseline models of each fold are tested on the CLOUDLAB dataset respectively so that we obtain the results averaged on 5 folds. The colors and signs have the same meanings as Fig. 4 and Fig. 9.

accuracy, CSSL algorithm performs slightly better than baseline models when predicting complex shapes of ice crystals which experienced microphysical processes in the cloud.

The evaluation on a 4-category NASCENT19 dataset shows that CSSL algorithm performs better than the baseline algorithm in general. The per-category analysis reveals that the CSSL algorithm demonstrates a better performance in predicting complex ice crystal shapes, achieving an average improvement of around 2.5 % in precision and an average improvement of around 3.5 % recall compared to a fully supervised algorithm. The generalization ability of CSSL algorithm is tested on a new 4-category dataset CLOUDLAB. The results reveal significant performance gaps between the CSSL algorithm based on the MoCo structure and the baseline models. On average, the CSSL algorithm outperforms the purely supervised model by 2.19 % across all training set sizes. It indicates a better generalization ability of the CSSL algorithm. It shows promising potential of CSSL about adapting to new datasets that are not used in training. The architecture of CSSL algorithm separates the feature learning process and the specific downstream task, which makes the model flexible to classifying new datasets through only fine-tuning on relatively few labeled examples. As the model has learnt the features of ice crystal in the upstream network, it can also adapt to data collected using different imaging devices such as VISSS (Maahn et al., 2024) and PIP&2DS (Jaffeux et al., 2022) with being fine-tuned on a small subset of new data, but the performance on such devices is required to be further evaluated in future studies.

However, this study uncovered several new challenges related to ice crystal classification, which we hope will be addressed in future research. The first issue is that the performance of classification models, no matter supervised or semi-supervised ones, is highly related to the distribution of the ice crystal categories. The overall accuracy is usually higher on the categories that share a large percentage in the training set. The unbalanced distribution of ice crystal categories is likely a common case in cloud environments. The future research could focus on addressing the unbalanced distribution issue. There are two potential approaches to address this issue. The first is to balance the dataset by downsampling the overrepresented categories. The second is to adjust the algorithm's prediction probabilities by artificially lowering the confidence for categories with larger shares and increasing it for the underrepresented categories.

The second issue is that the classification performance of models fine-tuned on small dataset (n < 2048) is unstable. A possible reason is that the unsupervised model is not well pre-trained due to a relatively small size of unlabeled dataset (33354 samples). It could be solved by collecting more ice crystal images and feeding to the algorithm during unsupervised pretraining. Moreover, the training set for unsupervised pre-training can be expanded as more ice crystal images will be collected, which can improve the quality of the features learned by incorporating the additional data. Expanding the scale of unsupervised pre-training enables the integration of datasets collected from different imaging probes, including (VISSS (Maahn et al., 2024) and PIP&2DS (Jaffeux et al., 2022)) possible. This approach makes it feasible to develop a foundational model for ice crystals that can be effectively transferred to downstream tasks, such as shape classification and the detection of component-scale ice crystals (Zhang et al., 2024).

Another direction for future research is to investigate the trade-off between backbone depth and performance. While our current backbone ResNet50 with 49 convolutional layers achieves good classification results, exploring the performance of shallower depths could potentially lead to more efficient architectures while maintaining classification accuracy. This direction

would be valuable for applications where computational resources are limited. Alternatively, testing with deeper backbones
could help us understand the upper limits of feature learning capacity for CSSL algorithm.

The CSSL (Contrastive Semi-Supervised Learning) algorithm introduces a novel approach for classifying large ice crystal datasets with minimal labeled data. By significantly reducing the need for manual labeling, it enables the training of classification models with only a small subset of the entire dataset. Through an analysis of the relationship between training set size, the number of categories, and the algorithm's performance, we determined that labeling 2,048 samples achieves an optimal balance
between model accuracy and manual labeling effort for both 19-category and 4-category classification tasks. Additionally, an evaluation of the CSSL algorithm's generalization ability demonstrates its potential to perform well on datasets collected under varying conditions, highlighting its adaptability and robustness across different scenarios.

**Appendix A: The loss values of unsupervised pre-training and supervised fine-tuning**

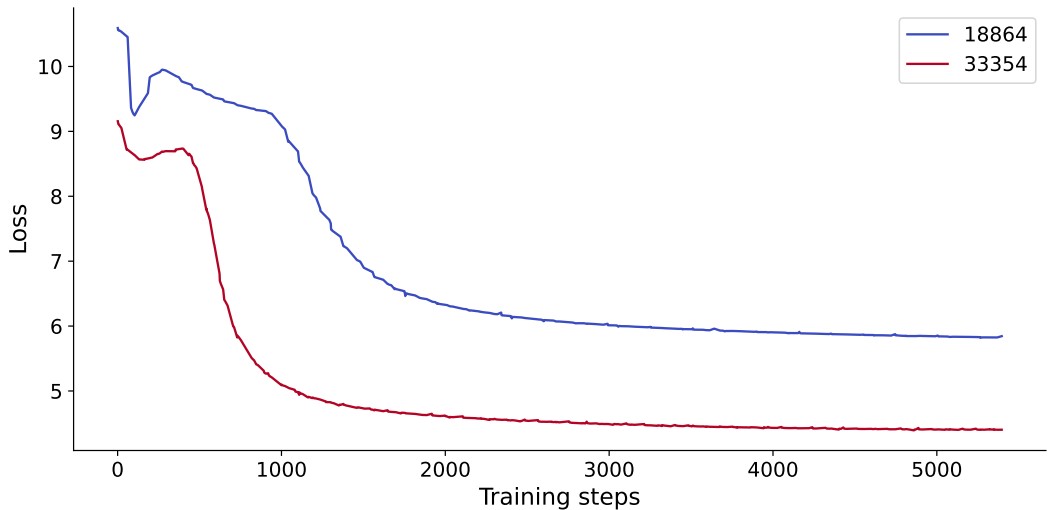

**Figure A1.** The loss values of unsupervised pre-training on the NASCENT19 dataset (blue line) and the whole NASCENT dataset (red line). The x-axis is the training steps.

The loss value shows how well a deep learning model is trained. When the loss value stop decreasing significantly, the model
can be considered as converged. The loss value of last step can indicate the model performance. In general, the smaller the value, the better the model can perform. To investigate why models trained on small datasets exhibit unstable performance, we analyzed the training loss for both Unsup-MoCo (Fig. A1) and Semisup-MoCo (Fig. A2).

We conducted unsupervised pre-training experiments using the NASCENT19 containing 18,864 samples and the complete NASCENT dataset containing 33,354 samples. It can be found that model pre-trained on NASCENT converged at a lower
loss value compared to the model pre-trained on the smaller NASCENT19 dataset. It indicates that increasing the size of

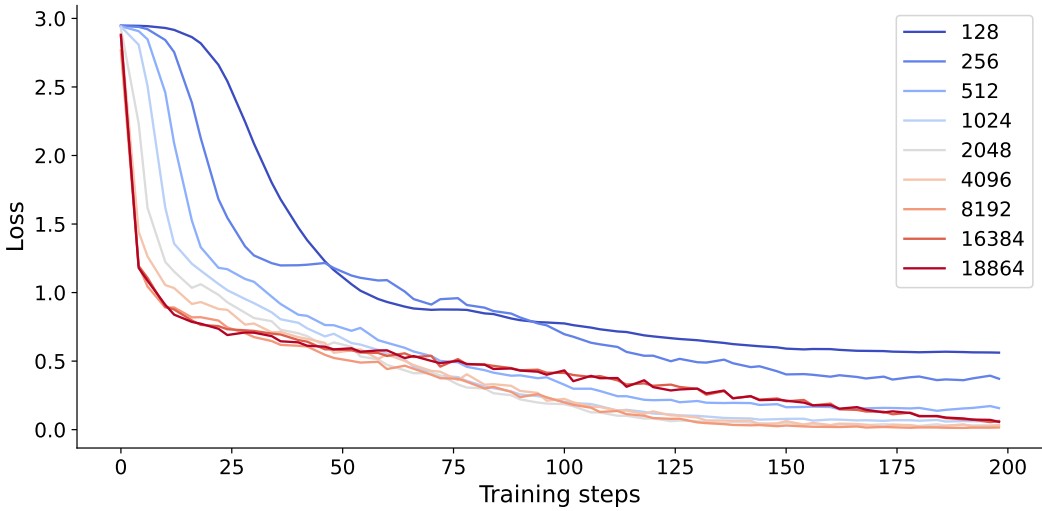

**Figure A2.** The loss values of supervised fine-tuning on the different size of training set (NASCENT19). The x-axis is the training steps.

unlabeled data in unsupervised pre-training leads to more effective feature learning, which in turn would improve downstream classification performance.

When examining the loss trending of semisup-MoCo models during the supervised fine-tuning, we observed that models trained on small datasets (128 and 256 samples) converged at higher loss values compared to those trained on larger datasets. It indicates the downstream network fine-tuned with small sizes dataset is not optimal, which could lead to unstable performance.

## Appendix B: Training times of unsupervised pre-training and supervised fine-tuning

The training times of models studied in this paper are listed in Table B1. The unsupervised pre-training phase required 66 minutes for the MoCo structure and 78 minutes for BYOL. For both the 19-category and 4-category classification tasks, the supervised fine-tuning phase of our models (Semisup-MoCo and Semisup-BYOL) consumed identical training time as their respective baseline models when trained on equivalent dataset sizes, hence, they are not displayed separately.

|  | Network | Size of training set (n) | Time (min) |
|---|---|---|---|
| Unsupervised pre-training | MoCo | 33354 | 66 |
|  | BYOL | 33354 | 78 |
| Supervised fine-tuning/Baseline | ResNet50 | 128 | 5 |
|  |  | 256 | 8 |
|  |  | 512 | 13 |
|  |  | 1024 | 29 |
|  |  | 2048 | 38 |
|  |  | 4096 | 44 |
|  |  | 8192 | 66 |
|  |  | 16384 | 85 |
|  |  | 18864 | 96 |

**Table B1.** The training times of models are listed, including the unsupervised pre-trained models, the supervised fine-tuned models with different training set sizes, and the baseline models. Note that for the supervised fine-tuned and baseline models, the listed training time represents a single fold of the cross-validation process.

*Code and data availability.* The dataset (https://doi.org/10.5281/zenodo.14696359, Chu et al., 2025a), models (https://doi.org/10.5281/zenodo.14793334, Chu et al., 2025b), and codes (https://doi.org/10.5281/zenodo.14795274, Chu et al., 2025c)

*Author contributions.* YC designed the algorithm. YC programmed all the codes related to training, evaluation and plotting. YC conducted all experiments. HZ, XL and JH closely and actively participated in discussion and give suggestions regarding the design of the algorithm and experiments. Both HZ and XL labeled the NASCENT19 and NASCENT20 dataset. YC and HZ collaborated to label the CLOUDLAB dataset. The manuscript is written by YC, with valuable input, review and discussion from HZ, XL and JH.

*Competing interests.* The authors declare that they have no conflict of interest.

*Acknowledgements.* The authors acknowledge the financial support from the European Research Council (ERC) under the European Union's Horizon 2020 research and innovation program (grant No. 101021272). YC also wants to express the heartfelt gratitude to Julie Pasquier

who did the initial classification of the NASCENT dataset, and NASCENT and CLOUDLAB team for their effort of collecting the data in

the field.

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
