# Peer review of "Exploring the effect of training set size and number of categories on ice crystal classification through a contrastive semi-supervised learning algorithm"

_EGUsphere, 2024_

## Referee Comment (RC2)

Determining the shape and morphology of ice particles is important because it provides a fingerprint of the cloud processes taking place. Chu et al. 2024 present a new ice particle shape recognition algorithm (contrastive semi-supervised learning algorithm - CSSL). CSSL is based on two steps, unsupervised training (upwards) and supervised fine-tuning (downwards). This method is new because the older methods are either fully supervised or based on feature analysis. In either case, a lot of manual hand labeling effort was required for training and evaluation. The aim of the study is to show that with the new CSSL algorithm it is possible to achieve similarly accurate shape discrimination with less manually labelled data. For this purpose, CSSL was trained and tested in three recent studies using images from a holographic imaging device.

Chu et al. 2024 highlight two points:

1. They show that their new algorithm requires only a small fraction of the particles to be labelled to achieve similar accuracy, saving many hours of labelling work.

2. They point out that the performance of classification models depends on the distribution of ice crystal categories and that this applies to all types of algorithms.

The article by Chu et al 2024 is new and shows that it is possible to distinguish particle shapes with less time spent labelling particles. The question of how many shape categories actually make sense remains open, but should be discussed in the future. The data set also seems a bit small and should definitely be expanded to include more different cases. Other shape categories could emerge, resulting in a different distribution of particle shapes.
The study by Chu et al 2024 is relevant to the community. It would be particularly helpful and useful if the algorithm could be made available to everyone on Github. After some minor improvements, this study can be published.

I have a few comments:

1. I would recommend explaining why you have explicitly chosen these 19 categories. For example, why the difference between columns and aged_columns is important and relevant, or why you don't have an extra class for sectorial plates. Explain how easy it is to include different particle shapes such as spherical rosettes?

2. Question about Figure 1: Are the sample images arranged randomly, or is there a reason for the arrangement? If it is random, I think it would be clearer to have similar groups together. I also wonder if the Column_and_aggregated image in the first row, second place, isn't more of a Lollypop_aged_aggregate, but that group doesn't exist.

3. In line 90 you write: 'Ageing' means that the ice crystals undergo processes such as riming, melting or sublimation. I would recommend that you consider whether

it is possible to distinguish between riming, melting or sublimation, and whether this consideration is worthwhile or not?

4.  In 3.1 you say that it is important for unsupervised algorithms to add additional samples to the dataset by converting images into different versions. I would recommend discussing up to what number of particles this first step is even necessary. Because if you have a lot of (e.g. more than a million) unlabelled particles. Isn't this more necessary for the second step, so that you have more labelled particles?

Typos and Image improvement:

Line 90: "aggregate".in → "aggregate" in

Line 92: "For example, in the case of riming, the image of aged crystals usually show a softly textured edges, which represents the supercooled droplets freezing on them."
Change either to: the image... shows ... edge
or: the images ...show ... softly textured edges

Line 116:  fatures → features

Line 134: sections→ sub-sections

Line 251: Fullstop missing

For Figures 6, 7 and 11: Increase the distance between number and percentage. In Sum Actual/Sum Predicted, the number and percentage are not easy to read. The contrast between black and dark blue is not so good, maybe change it to white on dark blue.

I really like this figure 8, it shows so well how your algorithm works.

---

## Author Comment (AC1)

Authors' responds to Referee Dr. Louis Jaffeux:

We would like to thank the reviewer for the thorough review of our manuscript and insightful feedback. These comments have significantly improved the quality of our work. In the following sections, we present the reviewer's comments (in black), our responses (in red), and the changes made in the revised manuscript (in blue). Please note that all line numbers in our responses correspond to those in the revised manuscript.

Comments

1. Public code and data:

   An associated GitHub repository or making the code and data public is highly encouraged. This article uses holographic imager data and could inspire researchers working with other image types, such as CCD imagers, optical array probes, or even 2D scattering probes, for which a wealth of hand-labeled datasets and trained algorithms already exist. The experiments could thus be easily reproduced with other data types and campaign datasets to validate the general conclusions on the CSSL algorithm.

   The data, code and models presented in this publication will be made available at: https://zenodo.org/communities/eth_zurich_iac_atmospheric_physics/

   The DOI links will be activated for public access upon acceptance of publication.

2. Some improvements can be made in the presentation of each model in the tables. Initially, the semi-supervised models are not straightforward to identify in Tables 4 and 5, which may carry over into further reading of the study. The two unsupervised models are listed in Table 4, while the two semi-supervised models are labeled as supervised models (which is technically true). The fact that both tables do not directly correspond to the experiments, due to the inclusion of the "Unsup" models and the classification of CSSL-generated models as "Sup," may be confusing for readers unfamiliar with the employed technique.

   Thank you for pointing this out. Following your comments, we first clarify that the whole network is semi-supervised in the manuscript, and explain why we call it 'semi-supervised'

   Figure 2

*Figure 1: The schematic of CSSL algorithm*

**L131-L133**

The downstream network adopts a traditional supervised image classification architecture, with a key distinction: the encoder is transferred from the upstream network. This transfer makes the whole network a semi-supervised learning approach, where the network uses both unsupervised knowledge from the upstream network and labeled image data from human.

**L211-L213**

The involvement of human knowledge (i.e. image labels) in the downstream network is the reason why we recognized the algorithm is semi-supervised.

We have renamed the models to reflect their types and structures. The type includes 'unsupervised (unsup)', 'semi-supervised (semisup)' and 'supervised (sup)'. 'Unsupervised' refers to the upstream network of CSSL, trained without labels. 'Semi-supervised' specifically refers to the classification stage of CSSL. 'Supervised' refers to purely supervised models. Structures include both MoCo and BYOL. In addition, we have renamed the weight initializations as 'dataset-type-structure' to clarify the source of the models' weights.

**L275-277**

In the rest of the paper, we will refer to a model by its type: unsupervised (unsup), supervised (sup) and semi-supervised (semisup); and its specific structure: MoCo and BYOL. The 'unsupervised' here represents the upstream network of CSSL algorithm. The 'semi-supervised' specifically refers to the classification stage of CSSL. 'Supervised' refers to purely supervised models.

**L279-L280**

the weights of both models are initialised with the weights of the respective structures pre-trained on the imagenet-1k dataset: IM1K-Unsup-MoCo and IM1K-Unsup-BYOL. In this paper, the weight initialisation will be refereed like 'dataset-type-structure'.

And then we modified the name of models and weight initialisations in Table 4 and 5 and in the manuscript according to our new naming systems. The unsupervised models are removed from tables since they are not the main object in the result section.

Table 4

| Name | Dataset | Weight initialization | Size of training set (n) | Categories (c) |
|---|---|---|---|---|
| Semisup-MoCo | NASCENT19 | NASCENT-Unsup-MoCo | [128, 256, 512, 1024, 2048, 4096, 8192, 16384, 18864] | 19 |
| Semisup-BYOL | NASCENT19 | NASCENT-Unsup-BYOL | [128, 256, 512, 1024, 2048, 4096, 8192, 16384, 18864] | 19 |
| Sup (Baseline) | NASCENT19 | IM1K-Unsup-MoCo | [128, 256, 512, 1024, 2048, 4096, 8192, 16384, 18864] | 19 |
| IceDetectNet | NASCENT19 | IM1K-Sup | 18864 | 19 |

*Table 1: The models trained and used for studying the effect of training set size.*

Table 4 caption

… "Semisup" represent the classification stage of CSSL which used both knowledge from unsupervised pre-training and image labels. "Sup" means that the models are purely supervised. "MoCo" and "BYOL" represent the encoders transferred from "Unsup-MoCo" or "Unsup-BYOL".

Table 5

| Name | Dataset | Weight initialization | Size of training set (n) | Categories (c) |
|---|---|---|---|---|
| Semisup-MoCo-4CAT | NASCENT19-4CAT | NASCENT-Unsup-MoCo | [128, 256, 512, 1024, 2048, 4096, 8192] | 4 |
| Semisup-BYOL-4CAT | NASCENT19-4CAT | NASCENT-Unsup-MoCo | [128, 256, 512, 1024, 2048, 4096, 8192] | 4 |
| Sup-4CAT (Baseline-4CAT) | NASCENT19-4CAT | IM1K-Unsup-MoCo | [128, 256, 512, 1024, 2048, 4096, 8192] | 4 |

*Table 2: The models trained and used for studying the effect of number of categories.*

3. Large error bars are found in Figures 4, 9, and 10 for small training sets. Additionally, the baseline model (fully supervised, with varying training set sizes) shows virtually the same performance as the two CSSL generated models. For the sake of transparency and setting realistic expectations for the paper, these limitations and the relative success of the experiments could be made more apparent in the abstract.

Thanks for your comments on the boxplot figures (Figure 4, 9 and 12). We did not have a clear and accurate elaboration of those results. We now improved our manuscript to explain the results more accurately. Firstly, we add explanations of each element in those boxplots. The central black lines in the boxes represent the median accuracy values of each 5-fold cross-validation experiment. The lower and upper limits of boxes are the 25% (the first) and 75% (the third) percentile (quantile) accuracy. Hence, the range of boxes displays the distribution of central 50% of accuracy values of each 5-fold cross-validation experiment, which means the average performance of each model. The error bars are the maximum and minimum accuracy values of each experiment, and the range of error bars shows the stability of each model.

Figure 4 caption

…The central black lines inside the boxes are the median values of accuracy of each 5-fold cross-validation experiment on the training set size. The lower limit and upper limit of boxes are the first quartile and the third quartile, respectively. The range of boxes shows the distribution of central 50\% accuracy values, which represents the average performance of each model. The error bars show the maximum and minimum accuracy values from each 5-fold cross-validation experiment…

After the additional explanations of boxplots, we would like to respond your comment points: The error bars range of CSSL is large when the training set sizes are small (especially for n=128 and n=256). We have already stated the finding and potential causes in L322, Section 5.1 for Figure 4. To make it clearer, we add more explanation:

L323-L329

One possible reason we concluded from checking the loss tendency during supervised fine-tuning on different sizes of dataset (Figure A2) is that the models fine-tuned on small sizes of dataset ($n< 2048$) is suboptimal compared to models fine-tuned on larger sizes, which would lead to unstable classification performance. Another possible reason we concluded from the loss value of unsupervised pre-training (Figure A1) is that the 33354 images may not be sufficient for optimizing the upstream network, which means the classification performance of CSSL algorithm could be further improved even when fine-tuning small size dataset if we pre-trained with more ice crystal images. We include the loss values in Appendix A.

We now add the same reason for Figure 9 in L388, Section 5.2.

L388-L389

The accuracy range of both the baseline and CSSL models trained on small datasets (n < 2048) are large. The reason is the same as the models fine-tuned on 19-category dataset.

In terms of Figure 10, it shows the trade-off between time saved on manual labeling and classification overall accuracy, which does not include the analysis of error bars. Therefore, we thought you may refer to Figure 12. As for Figure 12, we did not observe large error bar when the training size is small. At last, for the sake of transparency and setting realistic expectations, we add the content about large error bar in the abstract and the result section.

Abstract

L21-L23

Our analysis also reveals that both CSSL and purely supervised algorithms exhibit inherent instability when trained on small dataset sizes, …

Section 6

L439-L442

The second issue is that the classification performance of models fine-tuned on small dataset (n < 2048) is unstable. A possible reason is that the unsupervised model is not well pre-trained due

to a relatively small size of unlabeled dataset (33354 samples). It could be solved by collecting more ice crystal images and feeding to the algorithm during unsupervised pretraining.

In order to prove our arguments about large error bar, we add figures of loss values during training in the Appendix A. For proving that 33354 images may not be sufficient for unsupervised pre-training, we draw the correspond loss values.

[Figure]

*Figure 2: The loss of Unsup-MoCo during unsupervised pre-training use different size of unlabeled dataset*

The legend means the number of unlabeled image samples we used for unsupervised pre-training. We tried to pre-train the network use only NASCENT19 dataset (18864 samples) and the whole NASCENT dataset (33354 samples). It can be found that the loss values of the model pre-trained on 33354 samples are completely lower than the one pre-trained on 18864 sample, which indicates the network trained on 33354 samples converged better. Therefore, it is reasonable to assume the classification performance of models would be improved and more stable if we involve more data (>33354) during the stage unsupervised pre-training.

The models' performance become unstable during supervised fine-tuning because the volume of samples is not enough for models to converge. If we compare the loss values of Semisup-MoCo during the supervised fine-tuning in the following figure, we can clearly find that at the last step, the loss value is much higher when the training set size are 128 and 256, which reflect the fact that the models trained with 128 and 256 samples are not well converged.

[Figure]

*Figure 3: The loss of semisup-MoCo of different training set sizes.*

Appendix A.

The loss value shows how well a deep learning model is trained. When the loss value stop decreasing significantly, the model can be considered as converged. The loss value of last step can indicate the model performance. In general, the smaller the value, the better the model can perform. To investigate why models trained on small datasets exhibit unstable performance, we analyzed the training loss for both Unsup-MoCo (Figure A1) and Semisup-MoCo (Figure A2).

We conducted unsupervised pre-training experiments using the NASCENT19 containing 18,864 samples and the complete NASCENT dataset containing 33,354 samples. It can be found that model pre-trained on NASCENT converged at a lower loss value compared to the model pre-trained on the smaller NASCENT19 dataset. It indicates that increasing the size of unlabeled data in unsupervised pre-training leads to more effective feature learning, which in turn would improve downstream classification performance.

When examining the loss trending of semisup-MoCo models during the supervised fine-tuning, we observed that models trained on small datasets (128 and 256 samples) converged at higher loss values compared to those trained on larger datasets. It indicates the downstream network fine-tuned with small sizes dataset is not optimal, which could lead to unstable performance.

In terms of the close performance between baseline models and two CSSL models, we agreed that it is true when the training set sizes are larger 2048, because it can be found from the Figure 4 and 9 that the classification performance of semisup-MoCo is obviously better than the baseline when the training set size was lower than 2048. It shows the strong classification performance of MoCo based CSSL when the training set size used for fine-tuning is small (n<2048). In fact, the average classification accuracy of all models converges to a similar value when the training set size became larger than 2048, which reveals the existence of a threshold beyond which increasing training set sizes yields diminishing returns in model classification

performance improvement. We have already demonstrated it by Figure 5 and Figure 10 that there was a 'inflection point' for the trade-off between time spent on manual labelling and decreased overall accuracy.

Follow your advice, we add the statement of close performance among model when the training set size is large abstract for setting realistic expectations for the paper.

Abstract
L22-L23
…as well as the performance difference between them converges as the training set size exceeds 2048 samples…

4. Typos remain in the text. For example, in Section 3, the end of line 116, "useful fatures," should be corrected to "useful features."

Fixed, thanks!

---

## Author Comment (AC2)

Authors' responds to Anonymous Referee 3:

We would like to thank the reviewer for the thorough review of our manuscript and insightful feedback. These comments have significantly improved the quality of our work. In the following sections, we present the reviewer's comments (in black), our responses (in red), and the changes made in the revised manuscript (in blue). Please note that all line numbers in our responses correspond to those in the revised manuscript.

Comments

1. The computational requirements and training times for both CSSL and baseline models should be discussed, as these are relevant for practical implementation. The community would highly benefit if the code and data for the CSSL algorithm were made publicly available.

   The data, code and models presented in this publication will be made available at:
   https://zenodo.org/communities/eth_zurich_iac_atmospheric_physics/

   The DOI links will be activated for public access upon acceptance of publication.

   In terms of the computational requirements, we stated the GPU information in Section. 4.1. We now add more detailed information about the hardware environment

   L273-L274
   Beyond the GPU requirements, the algorithm requires a minimum computing environment consisting of a 4-core CPU and 16GB of system memory to operate.

   In terms of the training times of our algorithm, the information is now concluded in the Appendix B.

   Appendix B
   The training times of models studied in this paper are listed in Table B1. The unsupervised pre-training phase required 66 minutes for the MoCo structure and 78 minutes for BYOL. For both the 19-category and 4-category classification tasks, the supervised fine-tuning phase of our models (Semisup-MoCo and Semisup-BYOL) consumed equivalent training time as their respective baseline models when trained on equivalent dataset sizes, hence, they are not displayed separately.

| | Network | Size of training set (n) | Time (min) |
|---|---|---|---|
| Unsupervised pre-training | MoCo | 33354 | 66 |
| | BYOL | 33354 | 78 |
| Supervised fine-tuning/Baseline | ResNet50 | 128 | 5 |
| | | 256 | 8 |
| | | 512 | 13 |
| | | 1024 | 29 |
| | | 2048 | 38 |
| | | 4096 | 44 |
| | | 8192 | 66 |
| | | 16384 | 85 |
| | | 18864 | 96 |

*Table 1: The training times of models.*

2. Figure 1 would benefit from additional scale bars

The sample images shown in Figure 1 have been uniformly resized for demonstration purposes and do not reflect the actual physical dimensions of the ice crystals as captured by the imaging system. Therefore, it is hard to add scale bars while keeping the current figure unchanged. For detailed dimension information, Zhang et al. (2024) shows some sample images for each category with scale bars, which used the same dataset as this study.

L120-L121
A comprehensive collection of ice crystal examples can be found in the appendix of Zhang et al. (2024), where images of each distinct category are presented with scale bars indicating their actual dimensions.

3. Could you discuss the potential for transfer learning to other imaging systems that capture lower quality crystal images (e.g., those with coarser resolution) such as: VIZZZ, PIP/2DVD.

It would be highly interesting to test ice crystal images with a coarser resolution, keeping in mind that performance depends more on the number of pixels than on the actual size of the ice crystal. There are two potential approaches to transfer learning in this context. There are two potential approaches to transfer learning in this context. The first approach, as we mentioned in the conclusion section, involves incorporating the new dataset into our existing dataset and further training the pre-trained encoder within the upstream network, which can help us develop a foundational model for ice crystals that can be effectively transferred to downstream tasks. The second approach is also the main outcome of our research. It involves fine-tuning the pre-trained encoder directly on new datasets of varying sizes and categories. We now add some discussions about new imaging systems by following your comment.

L453-L456
…As the model has learnt the features of ice crystal in the upstream network, it can also adapt to data collected using different imaging devices such as VIZZZ (Maahn et al., 2024) and PIP&2DS (Jaffeux et al., 2022) with being fine-tuned on a small subset of new data, but the performance on such devices is required to be further evaluated in future studies…

…Expanding the scale of unsupervised pretraining enables the integration of datasets collected from different imaging probes, including (VIZZZ (Maahn et al., 2024) and PIP&2DS (Jaffeux et al., 2022)) possible…

4. Minor comments:

Fixed, thanks

5. While rerunning the analysis with fewer convolutional layers would be beyond the scope of this review, it would be valuable if you could elaborate on the choice of network architecture and its implications. In particular, the use of 49 convolutional layers raises questions about computational efficiency versus model performance. Could a shallower network potentially achieve similar results with reduced computational overhead?

The choice of 49 convolutional layers was based on our adoption of the ResNet-50 architecture, which was the common backbone across various computer vision tasks. A shallower encoder can improve computational efficiency while it may bring several concerns in the context of our task.

In the upstream network, we require an encoder with sufficient capacity to extract the features of ice crystals to perform contrastive learning, especially for those complex shapes. A deep encoder can extract a hierarchy of features, from basic edges and textures in the shallow layers to complex shape patterns in deeper layers (Zeiler and Fergus, 2014). Therefore, a shallower ResNet may not be sufficient for the upstream network to extract the detailed information of ice crystals such as the complicated structures of aged particles or aggregates. We now add the above argument as a background information in the introduction section.

…A deep network can extract a hierarchy of features, from basic edges and textures in the shallow layers to complex shape patterns in deeper layers (Zeiler and Fergus, 2014). In our task of learning the features of ice crystals, it is necessary to a sufficiently deep network to extract the detailed information such as the complicated structures of aged particles or aggregates…

Since the downstream network will directly use the encoder transferred from the upstream network, the architecture should keep the same as the encoder in the upstream network.

In addition, as shown in the Appendix B, despite using the ResNet-50 architecture, the computational overhead for unsupervised pre-training remains practical and efficient. The supervised fine-tuning process also demonstrates reasonable computational demands, particularly when working with smaller subsets of around 2048 samples

However, it is not necessary to conduct experiments with different depth of encoder. The choice of 49 convolutional layers in our implementation builds upon extensive prior research

on different visual tasks, demonstrating the effectiveness of ResNet50. In fact, the ResNet paper (He et al., 2016) had shown that ResNet-50, despite its greater depth, maintains comparable computational efficiency to the shallower ResNet-34 architecture. Specifically, ResNet-50 requires 3.8 GFLOPs compared to ResNet-34's 3.6 GFLOPs, while achieving a significant 3% reduction in error rate on ImageNet classification. We add a description of the effectiveness of ResNet50 in the manuscript.

L183-L185
…which was proved more efficient and effective than other variations of ResNet (He et al., 2016)…

Reference

Zeiler, M. D. and Fergus, R.: Visualizing and understanding convolutional networks, in: Computer Vision-ECCV 2014: 13th European Conference, Zurich, Switzerland, September 6-12, 2014, Proceedings, Part I 13, pp. 818-833, Springer, 2014.

---

## Author Comment (AC3)

Authors' responds to Anonymous Referee 2:

We would like to thank the reviewer for the thorough review of our manuscript and insightful feedback. These comments have significantly improved the quality of our work. In the following sections, we present the reviewer's comments (in black), our responses (in red), and the changes made in the revised manuscript (in blue). Please note that all line numbers in our responses correspond to those in the revised manuscript.
* * *
Comments

1. I would recommend explaining why you have explicitly chosen these 19 categories. For example, why the difference between columns and aged_columns is important and relevant, or why you don't have an extra class for sectorial plates. Explain how easy it is to include different particle shapes such as spherical rosettes?

The classification scheme in this study follows Zhang et al. (2024), who designed categories of ice crystals based on the basic habits and two microphysical processes (aggregate and aged). The main reason of choosing 19 categories is that the scheme is designed based on the NASCENT dataset in which 19 categories are identified. Following your comment, we add explanations about the reason we have 19 categories in the Data Section

L87-L95
…The scheme contains 7 basic habits identified by Pasquier et al. (2022b) from the ice crystal images collected and identified in mixed-phase clouds of Ny-Alesund during the NASCENT campaign (Pasquier et al., 2022a). When combined with two microphysical processes: aging and aggregation, these basic habits develop into 12 complex shape categories, after excluding combinations that were not feasible. Among the 7 basic habits, the "Plate" and "Column" formed due to deposition growth under different temperature and supersaturation conditions. "Lollipop" (Keppas et al., 2017) forms by a droplet freezing on a columnar ice crystal, or the columnar part is the result of depositional growth on a frozen droplet. "CPC (columns on capped-columns)" originated from cycling through the columnar and plate temperature growth regimes, during their vertical transport by in-cloud circulation (Pasquier et al., 2023). Ice crystals that are too small for shape determination are categorized as 'Small', while large crystals with indistinguishable shapes are categorized as 'Irregular' '. As for the two microphysical processes…

In terms of the detailed criteria for manually classifying ice crystal, it was comprehensively discussed in the appendix of Zhang et al. (2024).

The shape of ice crystals plays a crucial role in atmospheric processes, as mentioned in the introduction. Different shapes exhibit varying radiation properties, which ultimately influence Earth's radiative forcing. For example, Wendisch et al. (2007) had shown the upwelling irradiances reflected by cirrus cloud is significant shape dependence. Additionally, the shape of ice crystals affects precipitation formation and determines the type of precipitation that reaches the ground. For example, heavily rimed ice crystals such as aged column may leads to hail

eventually.

The scheme does not include other ice crystal shapes, such as sectorial plates, as these were not observed in the dataset collected during the NASCENT campaign. Their absence can be attributed to the temperature and supersaturation condition were not in favor of the formation of these shapes.

The classification scheme is flexible and can include other shapes like spherical rosettes. Researchers can adjust the categories after they determined the basic habits, and the microphysical processes based on the dataset they own. Besides, the CSSL algorithm is flexible and can easily adapts to new dataset that are not used in our original dataset. The architecture of CSSL algorithm separate the feature learning (unsupervised pre-training) and specific downstream task (classification), which make it flexible enough to handle new dataset as it can be fine-tuned on only a few labeled images, as shown in Section 5.2. We add some discussion about the flexibility of CSSL in the Section 6.

L446-L450
…It shows promising potential of adapting to new datasets that are not used in training. The architecture of CSSL algorithm separates the feature learning process and the specific downstream task, which makes the model flexible to classifying new datasets through only fine-tuning on relatively few labeled examples. As the model has learnt the features of ice crystal in the upstream network, it can also adapt to data collected using different imaging devices…

2. Question about Figure 1: Are the sample images arranged randomly, or is there a reason for the arrangement? If it is random, I think it would be clearer to have similar groups together. I also wonder if the Column_and_aggregated image in the first row, second place, isn't more of a Lollypop_aged_aggregate, but that group doesn't exist.

No, it was not arranged randomly. It was arranged according to the distribution of ice crystal categories in NASCENT19 dataset. In the left panel, the order of the Y-axis labels is based on the number of images in each category in the NASCENT19 dataset, from largest to smallest. To make it easier for readers to match the category names in the left panel with the corresponding images in the right panel, we have listed the images one by one in the order of the Y-axis labels from top to bottom. The correspondence of left panel and the right panel is as follows:

[Figure]

*Figure 1: The correspondence of the left panel and the right panel*

A Lollipop ice crystal forms by a droplet freezing on a columnar ice crystal or a frozen droplet forming a columnar part through deposition growth. The 'column_aged_aggregate' in the first row is a two heavily rimed 'column' aggregate together, which does not exhibit patterns of a freezing droplet.

3. In line 90 you write: 'Ageing' means that the ice crystals undergo processes such as riming, melting or sublimation. I would recommend that you consider whether it is possible to distinguish between riming, melting or sublimation, and whether this consideration is worthwhile or not?

   Thanks for commenting about the definition of ageing. We recognize the difficulty to distinguish between riming, melting and sublimation just from images within our dataset. Therefore, we used 'aged'/'ageing' as a category to include riming, melting or sublimation so that we can avoid misclassification of the CSSL algorithm. It also provides researchers the flexibility to further divide 'aged' category to riming, melting or sublimation when considering extra information such as the air temperature.

   In terms of the detailed criteria for manually classifying ice crystal, it was comprehensively discussed in the appendix of Zhang et al. (2024).

4. In 3.1 you say that it is important for unsupervised algorithms to add additional samples to the dataset by converting images into different versions. I would recommend discussing up to what number of particles this first step is even necessary. Because if you have a lot of (e.g. more than a million) unlabelled particles. Isn't this more necessary for the second step, so that you have more labelled particles?

Thanks for your comment about data augmentation. In contrastive unsupervised learning, data augmentation is principally essential, as we mentioned in Section 3.1. The algorithm requires different versions of the same image to learn, as it works by comparing these variations and measuring their similarities to compute the loss function. This need for multiple image versions exists regardless of the training set size, because the core principle of contrastive learning relies on the algorithm's ability to recognize that different augmented views of the same image should be considered similar, while views of different images should be treated as dissimilar.

In fact, during the supervised fine-tuning, the data augmentations also exist for increasing the number of samples. We now add this description in the manuscript to make it clearer.

L163-L167
During fine-tuning, data augmentation aims at expanding the size of our training dataset, which prevents overfitting (Shorten and Khoshgoftaar, 2019). The data augmentations during supervised fine-tuning include random cropping and random flipping. The input images in the downstream network are firstly resized to 256 × 256, and then they are cropped with area ranging from 60% to 100% randomly. Finally, they are randomly flipped before imputing into the CNN.

5. Typos.
   Fixed, thanks

**References**

Keppas, S., Crosier, J., Choularton, T., and Bower, K.: Ice lollies: An ice particle generated in supercooled conveyor belts, Geo-5 physical Research Letters, 44, 5222-5230, 2017.

Pasquier, J. T., David, R. O., Freitas, G., Gierens, R., Gramlich, Y., Haslett, S., Li, G., Schäfer, B., Siegel, K., Wieder, J., et al.: The Ny- Alesund aerosol cloud experiment (nascent): Overview and first results, Bulletin of the American Meteorological Society, 103, E2533- E2558, 2022a.

Pasquier, J. T., Henneberger, J., Ramelli, F., Lauber, A., David, R. O., Wieder, J., Carlsen, T., Gierens, R., Maturilli, M., and Lohmann, U.:Conditions favorable for secondary ice production in Arctic mixed-phase clouds, Atmospheric Chemistry and Physics, 22, 15 579-15 601, 2022b.

Pasquier, J. T., Henneberger, J., Korolev, A., Ramelli, F., Wieder, J., Lauber, A., Li, G., David, R. O., Carlsen, T., Gierens, R., et al.:Understanding the history of two complex ice crystal habits deduced from a holographic imager, Geophysical Research Letters, 50, e2022GL100 247, 2023.

Wendisch, M., Yang, P., and Pilewskie, P.: Effects of ice crystal habit on thermal infrared radiative properties and forcing of cirrus, Journal of Geophysical Research: Atmospheres, 112, 2007.

Zhang, H., Li, X., Ramelli, F., David, R. O., Pasquier, J., and Henneberger, J.: IceDetectNet: A rotated object detection algorithm for classifying components of aggregated ice crystals with a multi-label classification scheme, Atmospheric Measurement Techniques, 17, 7109-7128, 2024.

---

## Referee Report (RR1)

**Referee Report for :**

**Exploring the effect of training set size and number of categories on ice crystal classification through a contrastive semi-supervised learning algorithm (egusphere-2024-3160)**

The article presents a novel methodology aimed at enhancing deep learning techniques for recognizing ice crystal images. Analyzing the morphology of ice particles is crucial for understanding cloud ice formation, estimating their lifetime, and assessing their potential for precipitation. While convolutional neural networks (CNNs) have been successfully trained for this purpose, their effectiveness has been hindered by labor-intensive manual labeling, limited generalization capabilities, and strong dependence on the original training datasets, raising concerns about their objectivity.

To address these challenges, the new methodology employs a contrastive semi-supervised learning (CSSL) algorithm, which integrates both supervised and unsupervised learning approaches. This technique aims to reduce the need for extensive manual labeling while improving the objectivity and generalization of the trained models. CSSL models are tested against a fully supervised CNN trained on the same datasets of varying sizes in a thorough and well-structered set of experiments, however limited by the amount of available data.

The results indicate that while the CSSL approach is promising, its advantages over purely supervised CNN models are not immediately clear. Specifically, once the training set exceeds 2048 samples (a relatively small number) the performance of the fully supervised CNN matches that of the CSSL models. Additionally, the study acknowledges that classification performance still depends on the category distribution within the training dataset, classification of less-represented classes being less accurate. However, one CSSL model generalizes better than the fully supervised CNN. Therefore, the expectation is that with larger pre-training datasets, the CSSL approach will further enhance generalization and reduce manual labeling efforts even more, making it a more efficient and scalable solution for ice crystal classification.

Associated data sets, codes, and models are provided by the authors, ensuring reproductibility and adaptation for other data sets.

The quality of the article is good and the clarity has been improved to be more accessible to newcomers in deep learning. The methodology is valuable for researchers using in-situ cloud and precipitation particle imagers, making this study a useful contribution to the field. Given its strengths in methodology, accessibility, and reproducibility, I support publishing the article as is.

---

## Referee Report (RR2)

Thank you for this new, improved and very good version.

I have a few minor comments or typos.

Line 65 "we proposed" are you sure that you want to use past tense here?

Line 105: "For example, in the case of riming, the images of aged crystals usually show a softly textured edges, which represent the supercooled droplets freezing on them."

➔ delete "a"

Line 123: "were described in (Henneberger et al., 2023)"

➔ were described in Henneberger et al. (2023)

Line 124: Therefore, the data used from CLOUDLAB from are dominated

➔ delete the second from

For Fig. 6,7 and 11: Is it possible to increase the contrast of the blue percentages? Perhaps to a lighter blue?

Line 470 and Line 486: VIZZZ, the instrument is called VISSS, Video in-situ snowfall sensor

Line 486-487: ",including(VIZZZ(Maahn et al., 2024) andPIP&2DS(Jaffeux et al., 2022))possible"

➔ including  VISSS, (Maahn et al., 2024) and PIP&2DS, (Jaffeux et al., 2022) possible

---

## Author Response (AR2)

Authors' responds to Referee Dr. Louis Jaffeux:

We would like to thank the reviewer again for the final review report and feedback.

Authors' responds to Anonymous Referee #2:

We would like to thank the reviewer again for the final review report and feedback. We present the reviewer's comments (in black), our responses (in red).

1. Typos

   Fixed, thanks.

2. For Fig. 6,7 and 11: Is it possible to increase the contrast of the blue percentages? Perhaps to a lighter blue?

   We now changed the color map and the contrast of Fig. 6, 7 and 11 to improve the visibility.